# Hyperspherical Latents Improve Continuous-Token Autoregressive Generation

**Guolin Ke**
DP Technology
kegl@dp.tech

**Hui Xue**
Microsoft Research
xuehui@microsoft.com

 **Code**: `https://github.com/guolinke/SphereAR`

## Abstract

Autoregressive (AR) models are promising for image generation, yet continuous-token AR variants often trail latent diffusion and masked-generation models. The core issue is heterogeneous variance in VAE latents, which is amplified during AR decoding, especially under classifier-free guidance (CFG), and can cause variance collapse. We propose *SphereAR* to address this issue. Its core design is to constrain all AR inputs and outputs—*including after CFG*—to lie on a fixed-radius hypersphere (constant $\ell_2$ norm), leveraging hyperspherical VAEs. Our theoretical analysis shows that hyperspherical constraint removes the scale component (the primary cause of variance collapse), thereby stabilizing AR decoding. Empirically, on ImageNet generation, *SphereAR-H* (943M) sets a new state of the art for AR models, achieving FID 1.34. Even at smaller scales, *SphereAR-L* (479M) reaches FID 1.54 and *SphereAR-B* (208M) reaches 1.92, matching or surpassing much larger baselines such as MAR-H (943M, 1.55) and VAR-d30 (2B, 1.92). To our knowledge, this is the first time a pure next-token AR image generator with raster order surpasses diffusion and masked-generation models at comparable parameter scales.

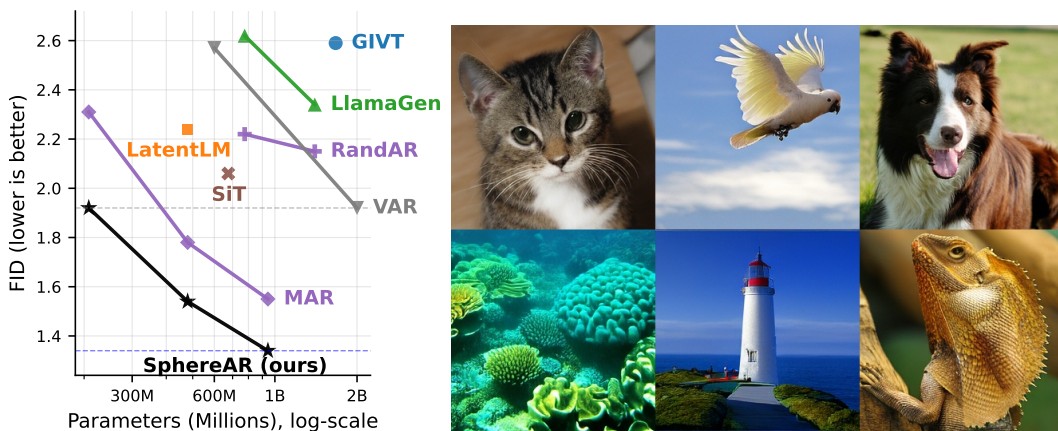

Figure 1: **Left:** FID vs. parameters on ImageNet 256×256 class-conditional generation, *SphereAR* attains lower FID with fewer parameters. **Right:** 256×256 samples generated by *SphereAR-L* (479M).

## 1 Introduction

Autoregressive (AR) models have achieved remarkable success in text (Radford et al., 2018; Brown et al., 2020) and have been extended to images (Esser et al., 2021; Yu et al., 2021), speech (Meng et al., 2024), video (Teng et al., 2025), and other modalities (Lu et al., 2025). Early multimodal AR systems discretized latents with vector quantization (VQ) (Gray, 1984; Van Den Oord et al., 2017); more recently, *continuous*-token AR dispenses with codebooks: a VAE (Kingma and Welling, 2013) emits token-level latents and the AR model predicts the next latent in continuous space (e.g., Gaussian mixtures (Tschannen et al., 2024) or diffusion objectives (Li et al., 2024a; Sun et al., 2024b)). Yet, when built on the same VAE latents, continuous-token AR models often trail latent

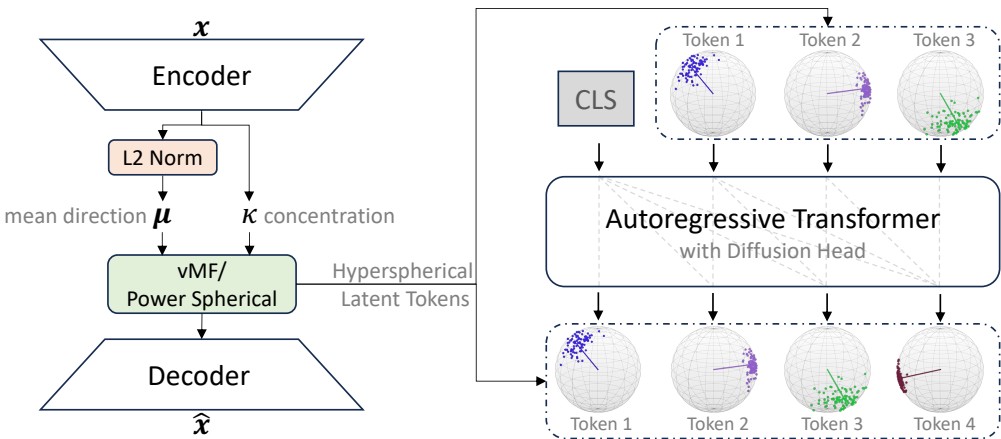

Figure 2: Overview of *SphereAR*. **Left:** A hyperspherical VAE (S-VAE) encodes raw data into a sequence of latent tokens constrained to a fixed-radius hypersphere $\mathbb{S}^{d-1}$. The encoder outputs a unit mean direction $\boldsymbol{\mu}$ and a concentration $\kappa$ that parameterize a von Mises–Fisher (vMF) or Power Spherical posterior. **Right:** A causal Transformer with a token-level diffusion head models the next-token distribution over the hyperspherical token sequence. At inference, the AR model's predictions, including CFG-rescaled ones, are projected back onto the fixed-radius hypersphere. The VAE decoder then reconstructs the image from the predicted hyperspherical latents.

diffusion and masked-generation models.[1] Prior analyses attribute this gap to variance pathologies during AR decoding (Sun et al., 2024b; Team et al., 2025): latent variances are heterogeneous across dimensions/tokens and are amplified due to exposure bias and classifier-free guidance (CFG) (Ho and Salimans, 2022), causing stepwise variance drift and collapse. Strengthening the KL term (Tschannen et al., 2024) or fixing a large variance (Sun et al., 2024b) improves stability but leaves the root cause intact: scale heterogeneity remains and can still drift during AR decoding with CFG.

We address this with a more principled solution: make all AR inputs and outputs *scale-invariant*. As illustrated in Fig. 2, the proposed *SphereAR* couples a hyperspherical VAE (S-VAE) (Davidson et al., 2018; De Cao and Aziz, 2020) with an autoregressive Transformer (Vaswani et al., 2017) and a token-level diffusion head (Li et al., 2024a). The S-VAE constrains each latent token to a fixed-radius hypersphere (constant $\ell_2$ norm), parameterizing only direction via a unit mean direction vector $\boldsymbol{\mu}$ and a scalar concentration $\kappa$. During training, the AR model consumes these hyperspherical latents under teacher forcing. During inference, AR model's predictions—including those after CFG rescaling—are projected back onto the fixed-radius hypersphere to remove the radial (scale) component. Thus, every signal provided to or produced by the AR model is $\ell_2$-normalized to the same radius. A concise theoretical justification supports these design choices, showing why scale-invariant inputs/outputs stabilize AR decoding and why a hyperspherical posterior is preferable to Gaussian alternatives.

Empirically, *SphereAR-H* (943M) sets a new state of the art for AR models on ImageNet $256\times256$ class-conditional generation, achieving FID 1.34. Even at smaller scales, *SphereAR-L* (479M) attains FID 1.54, outperforming comparably sized diffusion (DiT-XL/2, FID 2.27) and bidirectional masked-generation (MAR-L, FID 1.78) baselines, while matching MAR-H (943M, FID 1.55) with roughly half the parameters. At the base scale, *SphereAR-B* (208M) achieves FID 1.92, surpassing VAR-d20 (600M, FID 2.57) and the prior continuous-token AR model LatentLM-L (479M, FID 2.24), while matching VAR-d30 (2B, FID 1.92) with $\sim10\times$ fewer parameters. Ablations show that AR models with hyperspherical VAEs consistently outperform diagonal-Gaussian and fixed-variance $\sigma$-VAE (Sun et al., 2024b) baselines; moreover, applying post-hoc normalization to diagonal-Gaussian latents helps but still underperforms S-VAE. To our knowledge, this is the first time a pure next-token AR image generator with raster order surpasses diffusion and masked-generation models at comparable parameter scales.

---

[1]In this paper, "autoregressive" denotes token-by-token generation with unidirectional (causal) self-attention, excluding bidirectional masked/next-scale methods such as MaskGIT (Chang et al., 2022), MAR (Li et al., 2024a), and VAR (Tian et al., 2024). With the same VAE latents, Li et al. (2024a) report an AR model at FID 4.69 vs. 1.98 for MAR and 2.27 for DiT (Peebles and Xie, 2023).

## 2 RELATED WORK

**Image Tokenizers** A large body of work improves the performance of image tokenizers by enhancing reconstruction fidelity and semantic alignment. Typical ingredients include (i) refined training objectives (Esser et al., 2021; Yao et al., 2025; Yang et al., 2025), (ii) CLIP-aligned distillation for better text guidance (Peng et al., 2022; Qu et al., 2025), and (iii) various decoder improvements (Chen et al., 2025; Yang et al., 2025). These techniques are orthogonal to our approach and can be combined with it. In parallel, a complementary line of work targets the *quantization* mechanism itself—improving codebook utilization, training stability, and the rate-distortion tradeoff. Building on VQ-VAE (Van Den Oord et al., 2017), extensions include hierarchical VQ-VAE-2 (Razavi et al., 2019), residual/hierarchical quantization (Lee et al., 2022), and multi-codebook schemes (Li et al., 2024b). Some methods also adopt spherical or normalized feature geometry in the quantizer: for instance, ViT-VQGAN (Yu et al., 2021) normalizes latent features before computing codebook distances, and BSQ (Zhao et al., 2024) constructs binarized *spherical* latents for bit-efficient quantization.

By contrast, comparatively less work targets *continuous* image tokenizers tailored to autoregressive modeling. Most prior approaches follow latent diffusion practice (Peebles and Xie, 2023) and employ diagonal-Gaussian VAEs. GIVT (Tschannen et al., 2024) and LatentLM (Sun et al., 2024b) mitigate instability by inflating or fixing latent variance (e.g., $\beta$-VAE, $\sigma$-VAE), which helps but does not remove scale degrees of freedom. NextStep-1 (Team et al., 2025) instead normalizes Gaussian-posterior latents to a constant norm, achieving scale invariance. However, both our theoretical analysis and empirical results indicate that hyperspherical posteriors are preferable to post-hoc normalization of diagonal-Gaussian latents.

**Autoregressive Image Generation** Autoregressive image generation can be grouped into three families: *next-scale*, *next-set*, and *next-token* prediction. In *next-scale* prediction (e.g., VAR (Tian et al., 2024)), images are generated coarse-to-fine across resolutions; within each scale, context is modeled bidirectionally. In *next-set* prediction (also called masked generation; e.g., MaskGIT (Chang et al., 2022), MAR (Li et al., 2024a)), a single scale is used and a *set* of tokens is updated in parallel under bidirectional attention. In *next-token* prediction (e.g., VQGAN (Esser et al., 2021), LlamaGen (Sun et al., 2024a)), the model follows language-style sequence modeling: one token is predicted at a time with strictly unidirectional (causal) attention. We focus on next-token models because they align naturally with autoregressive language modeling and offer headroom for unified multimodal models.

A wide range of next-token variants has been explored: discrete tokens (Esser et al., 2021; Yu et al., 2021; Sun et al., 2024a) vs. continuous tokens (Tschannen et al., 2024; Sun et al., 2024b); raster order (Sun et al., 2024a; Tschannen et al., 2024) vs. randomized order (Pang et al., 2025; Yu et al., 2024); and more (Li et al., 2025). However, at comparable parameter scales, these models have often trailed next-set and next-scale approaches. A key reason is the variance collapse that emerges during autoregressive decoding (Sun et al., 2024b; Team et al., 2025). We address this by enforcing *scale-invariant* latents via hyperspherical VAEs, thereby removing scale degrees of freedom. Empirically, this yields substantial gains for sequential AR decoding, with performance that matches or surpasses state-of-the-art next-set and next-scale methods at comparable model budgets.

## 3 METHOD

We observe that with *discrete* tokens, next-token autoregressive (AR) models can outperform bidirectional masked-generation (MG) approaches. For example, LlamaGen-L (343M, FID 3.07; (Sun et al., 2024a)) vs. MaskGIT (207M, FID 4.02; (Chang et al., 2022)). A system-level study (Fan et al., 2024) further reports that, with discrete tokens, AR consistently achieves better FID than MG across model sizes from 166M to 3.1B. By contrast, with *continuous* tokens, MG is consistently stronger than AR. This divergence—discrete tokens thriving under AR while continuous tokens do not—motivates us to probe what truly differentiates the two. As illustrated in Fig.3, discrete tokens (Fig.3 a) are *normalized* on the probability simplex (components sum to 1), yielding *scale-invariant* inputs and outputs that stabilize AR decoding. In contrast, diagonal-Gaussian latents (Fig.3 b) are unconstrained and can destabilize multi-step AR decoding due to scale drift that compounds across steps. We hypothesize this scale sensitivity is the key issue, and therefore constrain continuous latents to a fixed-radius hypersphere to enforce a constant norm (Fig.3 c). This idea underpins *SphereAR*: a

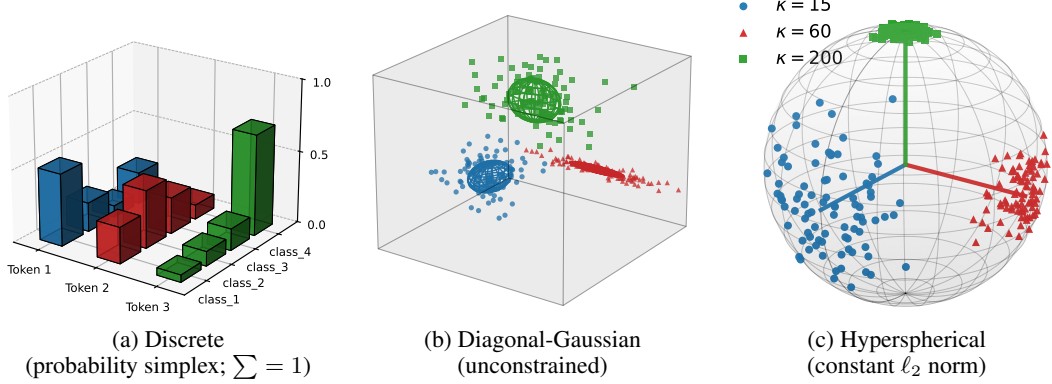

(a) Discrete
(probability simplex; $\sum = 1$)

(b) Diagonal-Gaussian
(unconstrained)

(c) Hyperspherical
(constant $\ell_2$ norm)

Figure 3: Visualization of token distributions. Each panel shows one token type, with three tokens in different colors. (a) Discrete tokens lie on the probability simplex and are intrinsically scale-invariant. (b) Diagonal-Gaussian latents are unconstrained in scale; despite a KL prior, per-dimension/token variances remain heterogeneous. (c) Hyperspherical latents constrain each token to a fixed norm (e.g., $\|\mathbf{z}\|_2 = R$), yielding scale-invariant representations. In practice, (a) and (c) are robust under AR decoding, whereas (b) is prone to scale drift and occasional variance collapse (e.g., with CFG).

hyperspherical VAE paired with a causal Transformer equipped with a token-level diffusion head; we detail these components below.

## 3.1 FROM VAE TO HYPERSPHERICAL VAE

A variational autoencoder (VAE) (Kingma and Welling, 2013) is widely used to compress raw data into a lower-dimensional latent vector. It consists of an encoder $q_\phi(\mathbf{z} \mid \mathbf{x})$ that parameterizes an approximate posterior over $\mathbf{z}$ and a decoder $p_\psi(\mathbf{x} \mid \mathbf{z})$ that reconstructs $\mathbf{x}$ from $\mathbf{z}$. We train the model by maximizing the evidence lower bound (ELBO):

$$\mathcal{L}(\phi, \psi; \mathbf{x}) = \mathbb{E}_{q_\phi(\mathbf{z}|\mathbf{x})}\big[\log p_\psi(\mathbf{x} \mid \mathbf{z})\big] - D_{\mathrm{KL}}\big(q_\phi(\mathbf{z} \mid \mathbf{x}) \,\|\, p(\mathbf{z})\big). \tag{1}$$

By default, both the prior $p(\mathbf{z})$ and the approximate posterior $q_\phi(\mathbf{z} \mid \mathbf{x})$ are parameterized as Gaussians with diagonal covariance; the prior is the isotropic standard Normal $\mathcal{N}(\mathbf{0}, \mathbf{I})$. Using the reparameterization trick, $\mathbf{z} = \boldsymbol{\mu}_\phi(\mathbf{x}) + \boldsymbol{\sigma}_\phi(\mathbf{x}) \odot \boldsymbol{\epsilon}$ with $\boldsymbol{\epsilon} \sim \mathcal{N}(\mathbf{0}, \mathbf{I})$, makes the sampling step differentiable so that gradients backpropagate from the decoder to the encoder.

With this diagonal-Gaussian posterior, the encoder's scale $\boldsymbol{\sigma}_\phi(\mathbf{x})$ is data-dependent and per-dimension, yielding *heterogeneous* variances across dimensions and tokens. This imbalance amplifies exposure bias and can trigger variance collapse in AR decoding, particularly under CFG (Sun et al., 2024b; Team et al., 2025).

**Hyperspherical VAE (S-VAE)** To fully address this issue, we remove the *scale* degree of freedom in the latent representation, rendering the AR model's inputs and outputs scale-invariant. Specifically, leveraging hyperspherical VAEs (S-VAEs) (Davidson et al., 2018; De Cao and Aziz, 2020), we constrain each latent token to lie on a fixed-radius hypersphere.

For each token, the S-VAE encoder parameterizes a *directional* posterior on the unit sphere by outputting a unit mean *direction* $\boldsymbol{\mu}_\phi(\mathbf{x}) \in \mathbb{S}^{d-1}$ (via $\ell_2$ normalization; $d$ is the latent dimension) and a nonnegative *concentration* $\kappa_\phi(\mathbf{x}) \in \mathbb{R}_{\geq 0}$. For notational convenience, let $\boldsymbol{\mu} = \boldsymbol{\mu}_\phi(\mathbf{x})$ and $\kappa = \kappa_\phi(\mathbf{x})$. S-VAE adopts a von Mises–Fisher (vMF) distribution (Davidson et al., 2018) for the directional approximate posterior:

$$q_\phi(\mathbf{u} \mid \mathbf{x}) = C_d(\kappa) \exp\big(\kappa \boldsymbol{\mu}^\top \mathbf{u}\big), \qquad \mathbf{u} \in \mathbb{S}^{d-1}, \tag{2}$$

where $C_d(\kappa) = \dfrac{\kappa^{\frac{d}{2}-1}}{(2\pi)^{\frac{d}{2}} I_{\frac{d}{2}-1}(\kappa)}$ is the normalizing constant and $I_\nu(\cdot)$ is the modified Bessel function of the first kind. Intuitively, $\boldsymbol{\mu}$ sets the preferred direction and $\kappa$ controls concentration: $\kappa = 0$ gives

the uniform distribution on $\mathbb{S}^{d-1}$, and larger $\kappa$ yields tighter mass around $\boldsymbol{\mu}$. Because $\boldsymbol{\mu}^\top \mathbf{u}$ is the cosine similarity on the sphere, the density in equation 2 increases as $\mathbf{u}$ aligns with $\boldsymbol{\mu}$.

We take the prior over directions to be uniform on the sphere, $p(\mathbf{u}) = \mathrm{Unif}(\mathbb{S}^{d-1})$, and use a fixed radius $R > 0$ (hyperparameter) so that $\mathbf{z} = R\,\mathbf{u}$ is fed to the decoder. The ELBO becomes

$$\mathcal{L}_{\text{S-VAE}}(\phi, \psi; \mathbf{x}) = \mathbb{E}_{q_\phi(\mathbf{u}|\mathbf{x})}\big[\log p_\psi(\mathbf{x} \mid \mathbf{z} = R\mathbf{u})\big] - D_{\mathrm{KL}}\big(q_\phi(\mathbf{u} \mid \mathbf{x}) \,\|\, p(\mathbf{u})\big). \qquad (3)$$

While vMF is principled for spherical latents, it can be less efficient due to the need for rejection sampling. For efficiency, we adopt the *Power Spherical* posterior (De Cao and Aziz, 2020) on $\mathbb{S}^{d-1}$,

$$q_\phi(\mathbf{u} \mid \mathbf{x}) \propto \big(1 + \boldsymbol{\mu}^\top \mathbf{u}\big)^\kappa, \qquad \mathbf{u} \in \mathbb{S}^{d-1}, \qquad (4)$$

which preserves spherical support and rotational symmetry yet admits a fully reparameterizable sampler *without* rejection sampling. For convenience, define the axial projection (cosine similarity) $c = \boldsymbol{\mu}^\top \mathbf{u} \in [-1, 1]$ with the affine transform $C = (c+1)/2 \in [0, 1]$. Under equation 4, the marginal of $C$ is a Beta distribution with parameters determined by $d$ and $\kappa$:

$$C \sim \mathrm{Beta}\Big(\alpha = \tfrac{d-1}{2} + \kappa, \ \beta = \tfrac{d-1}{2}\Big), \qquad \text{so that} \qquad c = 2C - 1. \qquad (5)$$

Sampling proceeds by drawing $C$ from the Beta and setting $c = 2C - 1$, then sampling a unit vector $\mathbf{v}_\perp$ uniformly in the tangent space orthogonal to $\boldsymbol{\mu}$ and composing

$$\mathbf{u} = c\,\boldsymbol{\mu} + \sqrt{1 - c^2}\,\mathbf{v}_\perp, \qquad (6)$$

optionally implemented via a Householder transform to align a reference basis with $\boldsymbol{\mu}$. This inverse-CDF construction yields low-variance, fully reparameterizable gradients and improved numerical stability; the spherical ELBO in equation 3 remains unchanged with $q_\phi$ taken as Power Spherical. In downstream autoregressive models, we keep the radius fixed and renormalize latent inputs/outputs back to $\|\mathbf{z}\|_2 = R$ (also after CFG rescaling) to remove scale degrees of freedom.

**Why Scale-Invariant Inputs and Outputs Matter in AR** We normalize each provisional next-token prediction by the radius-$R$ projection $N_R(\mathbf{z}) = R\,\mathbf{z}/\|\mathbf{z}\|_2$ onto the hypersphere. At a reference token on the sphere, the differential of $N_R$ is exactly the orthogonal projector onto the tangent space; thus, to first order, normalization removes radial (scale) perturbations and preserves only tangential (directional) ones. Consequently, composing normalization with the next-token predictor removes the radial (scale) component of the linearized one-step error *prior* to refeeding, so scale errors cannot accumulate across autoregressive steps. See Appendix A for the formal statement and proof.

**Limitations of Gaussian Posterior with Post-hoc Normalization** A tempting alternative to achieve scale invariance is to retain a diagonal-Gaussian posterior $q_\phi(\mathbf{z} \mid \mathbf{x})$ and normalize the sampled latents (via $N_R$), before feeding them to the decoder (henceforth "Gaussian+norm"). However, this choice is theoretically suboptimal: it optimizes a *strictly looser* variational bound than a spherical posterior (see Appendix B). Intuitively, the decoder discards radius by normalization, yet the ELBO still incurs an extra nonnegative *radial* KL term that does not arise with a hyperspherical posterior. By contrast, a hyperspherical posterior aligns the training objective with the constant-norm constraint and avoids this mismatch. Moreover, hyperspherical posteriors are axially symmetric about $\boldsymbol{\mu}$ and governed by a single concentration parameter $\kappa$, whereas Gaussian+norm induces a projected-normal (Angular Central Gaussian) directional law whose level sets are elliptical and generally not axially symmetric; this geometric mismatch makes it a poorer fit to purely directional structure (details in Appendix B). Empirically (Sec. 4.3), S-VAE outperforms Gaussian+norm, corroborating this analysis.

## 3.2 CONTINUOUS-TOKEN AUTOREGRESSIVE TRANSFORMER

Given an image $\mathbf{X} \in \mathbb{R}^{H \times W \times 3}$, S-VAE encodes it into a latent tensor $\mathbf{Z} \in \mathbb{R}^{h \times w \times d}$ with a *fixed per-token norm*: for every spatial location $(i, j)$, $\|\mathbf{Z}_{i,j}\|_2 = R$ (each $\mathbf{Z}_{i,j} \in \mathbb{R}^d$). For sequential autoregressive modeling, we flatten $\mathbf{Z}$ in *raster-scan* (row-major) order to obtain a sequence $\{\mathbf{z}_1, \ldots, \mathbf{z}_l\}$ of length $l = h\,w$, where $\mathbf{z}_k$ is simply the latent at the $k$-th position in row-major order.

We employ a causal (unidirectional) Transformer to model the conditional distribution of the next token in the flat sequence. At position $k - 1$, the model takes the prefix $\{\mathbf{z}_1, \ldots, \mathbf{z}_{k-1}\}$ as input and produces a hidden state $\mathbf{h}_{k-1} = f(\mathbf{z}_{<k}; \theta)$, where $\theta$ denotes the Transformer parameters. Optionally,

discrete class labels or text prompts are *prepended* as conditioning tokens to the prefix and included in the causal context.

To predict the next continuous token $\mathbf{z}_k$, we follow MAR (Li et al., 2024a) and attach a *token-level diffusion head*. Conditioned on $\mathbf{h}_{k-1}$, the head progressively transforms a simple prior (e.g., $\mathcal{N}(\mathbf{0}, \mathbf{I})$) into the data distribution of the next token $\mathbf{z}_k$.

We train the diffusion head with *Rectified Flow* (Lipman et al., 2022; Liu et al., 2022). Given a prior $\mathbf{z}_k^0 \sim \mathcal{N}(\mathbf{0}, \mathbf{I})$, target $\mathbf{z}_k^1 = \mathbf{z}_k$, and a continuous time $t \in (0, 1)$, we form the linear interpolation

$$\mathbf{z}_k^t = (1 - t)\,\mathbf{z}_k^0 + t\,\mathbf{z}_k^1. \tag{7}$$

The diffusion head, parameterized by $\omega$, takes the noisy interpolation $\mathbf{z}_k^t$, the scalar time $t$, and the condition $\mathbf{h}_{k-1}$ as inputs, and predicts a velocity, $\mathbf{v}_\omega(\mathbf{z}_k^t, t, \mathbf{h}_{k-1}) \in \mathbb{R}^d$. The training target is the flow velocity along the straight path, $\frac{d\mathbf{z}_k^t}{dt} = \mathbf{z}_k^1 - \mathbf{z}_k^0$, and the objective is mean-squared error:

$$\mathcal{L}_{\mathrm{RF}} = \mathbb{E}_{\mathbf{z}_k^0, \mathbf{z}_k^1, t}\left[\left\|\mathbf{z}_k^1 - \mathbf{z}_k^0 - \mathbf{v}_\omega(\mathbf{z}_k^t, t, \mathbf{h}_{k-1})\right\|_2^2\right]. \tag{8}$$

At inference, we initialize $\mathbf{z}_k^0 \sim \mathcal{N}(\mathbf{0}, \mathbf{I})$ at $t = 0$ and integrate the learned velocity field $\mathbf{v}_\omega(\mathbf{z}_k^t, t, \mathbf{h}_{k-1})$ up to $t = 1$ using $N$ uniform steps $\Delta = 1/N$ (e.g., explicit Euler):

$$\mathbf{z}_k^{t+\Delta} \leftarrow \mathbf{z}_k^t + \Delta\,\mathbf{v}_\omega(\mathbf{z}_k^t, t, \mathbf{h}_{k-1}). \tag{9}$$

After $N$ steps, we enforce the constant-norm constraint with a single projection onto the radius-$R$ hypersphere: $\mathbf{z}_k \leftarrow R\,\mathbf{z}_k^1/\|\mathbf{z}_k^1\|_2$. The resulting token $\mathbf{z}_k$ is then fed to the next AR step and ultimately to the VAE decoder. When using classifier-free guidance (CFG), the velocity at each step is obtained from a guided (rescaled) combination of conditional and unconditional predictions; we perform no intermediate normalization and apply a single constant-norm projection only after $N$ steps.

### 3.3 MODEL ARCHITECTURES

**S-VAE** Although VQGAN-style CNN backbones (Esser et al., 2021) are effective for latent VAEs, their throughput is limited by large convolutional activation maps. To improve efficiency without sacrificing quality, we adopt a *hybrid* backbone: the encoder uses a lightweight CNN stem with downsampling for *patchification*, followed by a stack of Transformer blocks; the decoder mirrors this with a Transformer stack that refines latent tokens and a lightweight CNN with upsampling for *unpatchification* and pixel reconstruction. This preserves the CNN's strong local inductive bias while leveraging the Transformer's efficient global modeling at token resolution, yielding a favorable speed–quality trade-off. As shown in Appendix D, the hybrid matches CNN baselines in quality while being about $2.6\times$ faster.

**Autoregressive Transformer** Following prior work (Sun et al., 2024a;b), we adopt a modern causal Transformer. Concretely, we use pre-norm Transformer blocks with RMSNorm (Zhang and Sennrich, 2019; Xiong et al., 2020), FlashAttention for efficient attention computation (Dao et al., 2022), and SwiGLU feed-forward layers (Shazeer, 2020). For image positional encoding, we employ 2D rotary embeddings (RoPE) (Su et al., 2024) applied in raster-scan order. All self-attention is strictly unidirectional (causal mask). For the diffusion head, we follow MAR (Li et al., 2024a) and use an MLP architecture.

## 4 EXPERIMENTS

We evaluate *SphereAR* on ImageNet-1K (Deng et al., 2009) class-conditional generation of a resolution of 256×256, comparing against previous strong baselines. Beyond end-to-end comparisons, we include targeted studies to substantiate our design choices, focusing on the following questions: (1) **S-VAE vs. diagonal-Gaussian:** Does S-VAE outperform diagonal-Gaussian VAEs for continuous-token AR? (2) **Post-hoc normalization:** If we apply $\ell_2$ normalization to latents from a diagonal-Gaussian VAE, how does it compare with S-VAE? (3) **Component contributions:** Which parts of S-VAE drive the gains—(i) the hyperspherical posterior, (ii) normalization applied to the VAE decoder input, or (iii) normalization applied to AR inputs/outputs?

### 4.1 IMPLEMENTATION DETAILS

**S-VAE**  We adopt a Power Spherical De Cao and Aziz (2020) directional posterior with latent dimensionality $d = 16$ and fix the radius to $R = \sqrt{d}$. Complete setting for the S-VAE's backbone is provided in Appendix D. We train S-VAE from scratch on ImageNet-1K (Deng et al., 2009) with random-crop augmentation, optimizing a weighted sum of ELBO (reconstruction + KL), perceptual (Johnson et al., 2016; Zhang et al., 2018), and adversarial (Isola et al., 2017) losses. Optimization uses AdamW (Kingma, 2014; Loshchilov and Hutter, 2017) for 100 epochs (batch size 256, learning rate $1 \times 10^{-4}$, $\beta = (0.9, 0.95)$, weight decay 0.05).

**Autoregressive Transformer**  Following MAR (Li et al., 2024a), we instantiate three model sizes for *SphereAR*. *SphereAR-B* uses 24 Transformer blocks (hidden size 768) and a diffusion head with 6 feed-forward blocks (hidden size 768). *SphereAR-L* uses 32 Transformer blocks (hidden size 1024) and a diffusion head with 8 feed-forward blocks (hidden size 1024). *SphereAR-H* uses 40 Transformer blocks (hidden size 1280) and a diffusion head with 12 feed-forward blocks (hidden size 1280). As in MAR, we employ multiple class-conditioning tokens (16 in our models vs. 64 in MAR) and apply class-token dropout with probability 0.1 during training to enable classifier-free guidance (CFG) at inference. Models are trained on ImageNet-1K for 400 epochs using S-VAE latents with AdamW (batch size 512, $\beta = (0.9, 0.95)$, weight decay 0.05), a cosine learning-rate schedule with 20k linear warmup steps and peak learning rate $3 \times 10^{-4}$, and an exponential moving average (EMA) of weights with decay 0.9999. Under these settings, *SphereAR-B*, *SphereAR-L* and *SphereAR-H* contain $\sim$208M, $\sim$479M and $\sim$943M parameters, respectively.

**Inference Settings**  For next-token prediction we integrate the learned velocity field with a fixed-step Euler scheme (100 steps). We use the linear CFG schedule from MAR. We enable a KV cache to improve autoregressive decoding efficiency.

### 4.2 IMAGE GENERATION RESULT

We report Fréchet Inception Distance (FID) (Heusel et al., 2017) as the primary metric, computed on 50k samples drawn with a fixed random seed using the ADM evaluation code (Dhariwal and Nichol, 2021). The optimal CFG scale is determined through a sweep with a step size of 0.1. Following MAR (Li et al., 2024a), we additionally report Inception Score (IS) (Salimans et al., 2016) and Precision/Recall (Pre./Rec.) (Kynkäänniemi et al., 2019).

From the results summarized in Table 1, we observe: (1) *SphereAR-H* (943M) achieves state-of-the-art FID **1.34**, outperforming VAR-d30 (2B, **1.92**) and MAR-H (943M, **1.55**). (2) *SphereAR is parameter-efficient.* At large scale, *SphereAR-L* (479M) matches MAR-H (943M, **1.55**) with roughly half the parameters. Even at the base scale, *SphereAR-B* (208M) reaches FID **1.92**, outperforming 2B-parameter VAR, diffusion baselines (DiT and SiT), prior continuous-token AR models (GIVT and LatentLM-L), and larger discrete AR models (LlamaGen and RandAR). (3) *Hyperspherical latents are critical.* The key difference from LatentLM is the latent parameterization—fixed-variance diagonal-Gaussian vs. hyperspherical. The large gap (*SphereAR-L*: **1.54** vs. LatentLM-L: **2.24**) indicates that constant-norm, directional latents are crucial for high-quality AR decoding.

Overall, *SphereAR* delivers a scale-invariant AR model that sets the best reported FID with far fewer parameters and outperforms diffusion, masked-generation, and next-scale baselines. Appendix K shows qualitative results.

### 4.3 ABLATION STUDY

All variants in this ablation use the same model backbone and training/evaluation configuration; only the VAE *objective/posterior* differs. To reduce compute, each VAE is trained on ImageNet for 50 epochs. For the AR stage we use the *SphereAR-L* backbone, trained for 50 epochs with a constant learning rate $1 \times 10^{-4}$ and batch size 256; all other settings follow Sec. 4.2.

**S-VAE vs. Diagonal-Gaussian**  The core design of *SphereAR* is to constrain latents on a hypersphere via S-VAE. We compare it with prior VAEs. In particular, we evaluate three settings: (1) *Diagonal-*

Table 1: Overall comparison on ImageNet 256×256 class-conditional generation. Abbreviations: AR = next-token (causal) autoregression; Mask. = masked generation (next-set); N.S. = next-scale; Diff. = diffusion. An asterisk (*) indicates models trained at 384×384 and evaluated at 256×256 by resizing.

| Model | Type | Order | #Params | #Epochs | FID↓ | IS↑ | Pre.↑ | Rec.↑ |
|---|---|---|---|---|---|---|---|---|
| *Discrete Tokens* | | | | | | | | |
| VQGAN (Esser et al., 2021) | AR | raster | 1.4B | 240 | 5.20 | 280.3 | - | - |
| ViT-VQGAN (Yu et al., 2021) | AR | raster | 1.7B | 240 | 3.04 | 227.4 | - | - |
| LlamaGen-L (Sun et al., 2024a) | AR | raster | 343M | 300 | 3.07 | 256.1 | 0.83 | 0.52 |
| LlamaGen-XL* (Sun et al., 2024a) | AR | raster | 775M | 300 | 2.62 | 244.1 | 0.80 | 0.57 |
| RandAR-L (Pang et al., 2025) | AR | random | 343M | 300 | 2.55 | 288.8 | 0.81 | 0.58 |
| RandAR-XL (Pang et al., 2025) | AR | random | 775M | 300 | 2.22 | 314.2 | 0.80 | 0.60 |
| RAR-B (Yu et al., 2024) | AR | hybrid | 261M | 400 | 1.95 | 290.5 | 0.82 | 0.58 |
| RAR-L (Yu et al., 2024) | AR | hybrid | 461M | 400 | 1.70 | 299.5 | 0.81 | 0.60 |
| MaskGIT (Chang et al., 2022) | Mask. | random | 227M | 300 | 4.02 | **355.6** | 0.78 | 0.50 |
| MAGVIT-v2 (Yu et al., 2023) | Mask. | random | 307M | 270 | 1.78 | 319.4 | - | - |
| VAR-d20 (Tian et al., 2024) | N.S. | - | 600M | 350 | 2.57 | 302.6 | 0.83 | 0.56 |
| VAR-d30 (Tian et al., 2024) | N.S. | - | 2B | 350 | 1.92 | 323.1 | 0.82 | 0.59 |
| *Continuous Tokens* | | | | | | | | |
| LDM-4 (Rombach et al., 2022) | Diff. | - | 400M | - | 3.60 | 247.7 | 0.87 | 0.48 |
| DiT-XL/2 (Peebles and Xie, 2023) | Diff. | - | 675M | 400 | 2.27 | 278.2 | 0.83 | 0.57 |
| SiT-XL/2 (Ma et al., 2024) | Diff. | - | 675M | 400 | 2.06 | 277.5 | 0.83 | 0.59 |
| GIVT (Tschannen et al., 2024) | AR | raster | 1.67B | 500 | 2.59 | - | 0.81 | 0.57 |
| LatentLM-L (Sun et al., 2024b) | AR | raster | 479M | 400 | 2.24 | 253.8 | - | - |
| MAR-B (Li et al., 2024a) | Mask. | random | 208M | 800 | 2.31 | 281.7 | 0.82 | 0.57 |
| MAR-L (Li et al., 2024a) | Mask. | random | 479M | 800 | 1.78 | 296.0 | 0.81 | 0.60 |
| MAR-H (Li et al., 2024a) | Mask. | random | 943M | 800 | 1.55 | 303.7 | 0.81 | 0.62 |
| *SphereAR-B* (Our) | AR | raster | 208M | 400 | 1.92 | 277.8 | 0.81 | 0.61 |
| *SphereAR-L* (Our) | AR | raster | 479M | 400 | 1.54 | 295.9 | 0.80 | 0.63 |
| *SphereAR-H* (Our) | AR | raster | 943M | 400 | **1.34** | 300.0 | 0.80 | 0.64 |

*Gaussian (β-VAE).* A standard diagonal-Gaussian posterior trained with an up-weighted KL term.[2] We sweep four KL weights {0.01, 0.04, 0.08, 0.16}, denoted `G-01`, `G-04`, `G-08`, and `G-16`. (2) *σ-VAE (fixed variance).* Following LatentLM (Sun et al., 2024b), we fix the posterior scale with a *fixed, non-learned* scalar $\sigma \sim \mathcal{N}(0, C_\sigma)$, and sample $\mathbf{z} = \boldsymbol{\mu}_\phi(\mathbf{x}) + \sigma\,\boldsymbol{\epsilon}$ with $\boldsymbol{\epsilon} \sim \mathcal{N}(\mathbf{0}, \mathbf{I})$. We sweep $C_\sigma \in \{0.1, 0.2, 0.5\}$ (denoted `F-01`, `F-02`, `F-05`). (3) *S-VAE (hyperspherical).* A Power Spherical posterior on $\mathbb{S}^{d-1}$ with a KL-weight sweep {0.001, 0.004, 0.008}, denoted `S-01`, `S-04`, `S-08`.

Fig. 4 (left) plots FID versus CFG for these posteriors. We observe: (1) *S-VAE is consistently best and most stable across CFG.* In particular, `S-04` attains the lowest FID and `S-08` is a close second. (2) *Stronger regularization helps diagonal-Gaussian but saturates.* Increasing $\beta$ (or $C_\sigma$) improves the curves, yet they become unstable at larger CFG and remain below S-VAE. (3) *Fixed variance offers no advantage.* σ-VAE variants achieve performance on par with standard diagonal-Gaussian VAEs (e.g., `F-02` vs. `G-04`; `F-05` vs. `G-08`), indicating that fixing the posterior scale does not help.

Overall, the above ablation isolates the tokenizer's role: hyperspherical VAEs yield the most robust and best final AR performance.

**Post-hoc Normalization on Diagonal-Gaussian**   As discussed in Sec. 3.1, a simple alternative to achieve scale invariance is to apply a post-hoc normalization to latents from a diagonal-Gaussian posterior. We therefore run an empirical ablation. Starting from `G-x` models, we project each latent

---

[2]Most prior VAEs compute the KL by taking a `sum` over spatial-channel dimensions ($h \times w \times d$) and then a batch `mean`. We instead take a `mean` over *all* elements (batch and spatial-channel), which lowers the numerical KL value; to match the effective regularization strength, we therefore use larger KL weights (our $10^{-2}$ roughly matches a prior $2 \times 10^{-6}$).

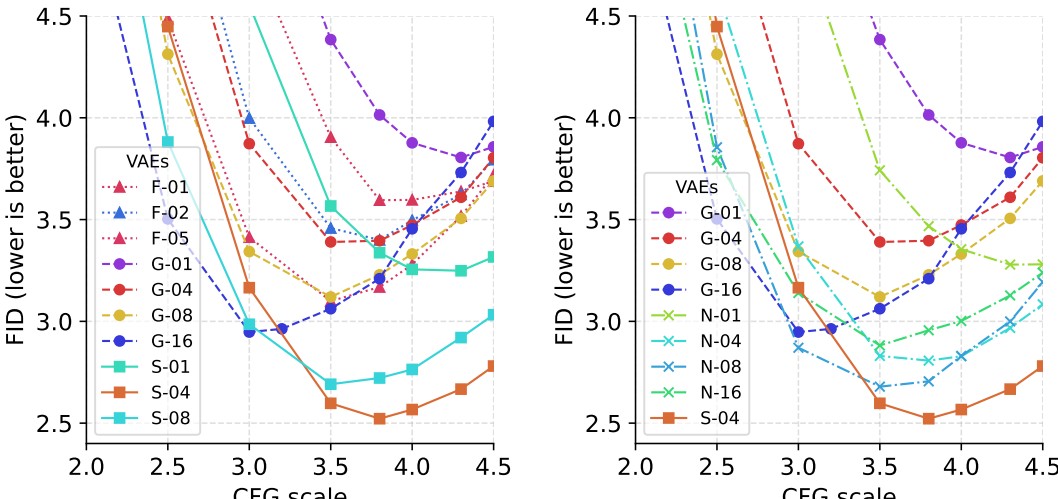

Figure 4: Impact of VAE variants on generation performance (FID vs. CFG). All variants share the same backbone and training/evaluation setup; only the VAE objective/posterior differs. **Left:** diagonal-Gaussian with enlarged KL weight (`G-01/04/08/16`), $\sigma$-VAE with fixed scale (`F-01/02/05`), and S-VAE with a Power Spherical posterior (`S-01/04/08`). **Right:** additionally includes diagonal-Gaussian with post-hoc normalization (`N-01/04/08/16`).

Table 2: Ablation of normalization (applied to the VAE decoder and AR) and posterior family.

| No. | Norm. on VAE Decoder | Norm. on AR | Posterior | FID↓ | IS↑ |
|-----|:--------------------:|:-----------:|-----------|------|------|
| 1 | ✗ | ✗ | Gaussian | 2.97 | 240.2 |
| 2 | ✓ | ✗ | Gaussian | 2.89 | 254.3 |
| 3 | ✓ | ✓ | Gaussian | 2.68 | 257.3 |
| 4 | ✓ | ✓ | **Spherical** | **2.52** | **258.4** |

onto the radius-$R$ hypersphere via $R\,\mathbf{z}/\|\mathbf{z}\|_2$, yielding `N-01`, `N-04`, `N-08`, and `N-16`. Fig. 4 (right) plots FID versus CFG for these variants.

From these results, we observe: (1) *Post-hoc normalization helps.* Each `N-x` improves over its `G-x` counterpart and is more stable at higher CFG scales, supporting our motivation that scale-invariant inputs/outputs stabilize AR decoding. (2) *S-VAE remains the best.* `S-04` outperforms the best post-hoc–normalized Gaussian (`N-08`). This aligns with our analysis: Gaussian with post-hoc normalization optimizes a *strictly looser* variational bound than the hyperspherical ELBO (Appendix B) and induces a non-axially symmetric directional law on $\mathbb{S}^{d-1}$.

**S-VAE's Component Contributions** The above ablations indicate that both the normalization on latent tokens and the hyperspherical posterior are important. Because the normalization can affect two interfaces—the VAE decoder's input and the AR model's inputs/outputs—we further isolate their effects by conducting a variant that normalizes only the VAE decoder's input. As summarized in Table 2, *normalization applied to the AR inputs and outputs is more critical*. In particular, normalizing only the VAE decoder's input yields a modest gain (FID $2.97 \rightarrow 2.89$, compare No. 1 with No. 2); whereas additionally normalizing AR inputs/outputs produces a larger improvement (FID $2.89 \rightarrow 2.68$, compare No.2 with No. 3). This finding supports our analysis: the AR pathway re-feeds tokens step by step, so scale errors would otherwise accumulate. In contrast, the VAE decoder consumes its input in a single pass and does not induce such cascading scale drift. Finally, adopting the Spherical posterior (No. 4) provides a further significant boost (FID $2.68 \rightarrow 2.52$), confirming that aligning the posterior with constant-norm geometry is beneficial.

## 5 CONCLUSION

To address variance collapse in continuous-token AR models, we propose *SphereAR*, whose core idea is to make all AR inputs and outputs scale-invariant. Concretely, it consists of (1) a hyperspherical VAE (S-VAE) that produces latent tokens constrained to a fixed-radius hypersphere; and (2) an autoregressive Transformer with a token-level diffusion head modeling the next-token distribution over hyperspherical latents. During AR training and inference, all inputs and outputs—including those after CFG rescaling—are normalized onto this hypersphere. Our theoretical analysis demonstrates that scale-invariant inputs and outputs are critical to AR modeling. On ImageNet class-conditional generation, *SphereAR-H* (943M) achieves FID 1.34 and *SphereAR-L* (479M) achieves FID 1.54, surpassing prior diffusion and masked-generation baselines. Ablations point to two key factors: constant-norm AR refeeding and a hyperspherical posterior. With both, S-VAE is best, exceeding diagonal-Gaussian, $\sigma$-VAE, and even diagonal-Gaussian with post-hoc normalization.

**Future work** While our results substantiate the motivation and design choices of *SphereAR*, several extensions would further strengthen this work: (i) exploring Riemannian Flow Matching (RFM) (Chen and Lipman, 2023), which may better align with hyperspherical latent geometry since trajectories of RFM remain on the hypersphere; and (ii) extending *SphereAR* to more datasets and multimodal applications. We leave these to future work.

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

## A   FIRST-ORDER STABILITY OF RADIUS PROJECTION IN AR

Let $N_R : \mathbb{R}^d \setminus \{\mathbf{0}\} \to \mathbb{S}_R^{d-1}$ be the radial projection $N_R(\mathbf{z}) = R\,\mathbf{z}/\|\mathbf{z}\|_2$, where $\mathbb{S}_R^{d-1} = \{\mathbf{z} \in \mathbb{R}^d : \|\mathbf{z}\|_2 = R\}$. All linearizations are taken at reference tokens $\bar{\mathbf{z}}_k = R\,\bar{\mathbf{u}}_k$ with $\|\bar{\mathbf{u}}_k\|_2 = 1$.

Let $g$ denote the *pre-normalization next-token map* implemented by our model: given the prefix $\mathbf{z}_{<k}$, the causal Transformer produces a condition $\mathbf{h}_{k-1}$, and the diffusion head returns a provisional latent $\tilde{\mathbf{z}}_k = g(\mathbf{z}_{<k}) \in \mathbb{R}^d$, which is then projected by $N_R$. We assume $g$ is *continuously differentiable* in a neighborhood of $\bar{\mathbf{z}}_{<k}$ (i.e., $C^1$: its Jacobian $\partial g/\partial \mathbf{z}_{<k}$ exists and varies continuously there), which holds for our architecture with smooth activations and fixed-step explicit ODE solvers. [3]

**Lemma 1 (Jacobian is the tangent-space projector).**   For $\|\bar{\mathbf{z}}_k\|_2 = R$,

$$\mathrm{D}N_R(\bar{\mathbf{z}}_k) \;=\; \mathbf{P}_k \;\coloneqq\; \mathbf{I} - \frac{\bar{\mathbf{z}}_k \bar{\mathbf{z}}_k^\top}{R^2}. \tag{10}$$

Moreover, $\mathbf{P}_k^\top = \mathbf{P}_k$, $\mathbf{P}_k^2 = \mathbf{P}_k$, $\mathbf{P}_k \bar{\mathbf{z}}_k = \mathbf{0}$, and $\mathbf{P}_k \mathbf{v} = \mathbf{v}$ for all $\mathbf{v} \in T_{\bar{\mathbf{z}}_k}\mathbb{S}_R^{d-1} = \{\mathbf{v} : \bar{\mathbf{z}}_k^\top \mathbf{v} = 0\}$; in particular, $\|\mathbf{P}_k\|_2 = 1$. *Proof.* Differentiate $N_R(\mathbf{z}) = R\,\mathbf{z}\,\|\mathbf{z}\|_2^{-1}$ and evaluate at $\bar{\mathbf{z}}_k$.

**Lemma 2 (First-order scale invariance).**   For any small $\Delta\mathbf{z}$,

$$N_R(\bar{\mathbf{z}}_k + \Delta\mathbf{z}) = \bar{\mathbf{z}}_k + \mathbf{P}_k\,\Delta\mathbf{z} + o(\|\Delta\mathbf{z}\|), \tag{11}$$

so radial derivatives vanish and tangential derivatives are preserved (eigenvalues 0 and 1). *Proof.* First-order Taylor expansion with Lemma 1.

**Proposition (One-step AR refeeding error, linearized).**   Let $g$ be the (unnormalized) next-token predictor and define $\tilde{\mathbf{z}}_k = g(\mathbf{z}_{<k})$, $\mathbf{z}_k = N_R(\tilde{\mathbf{z}}_k)$, and $F_k = N_R \circ g$. Linearizing at $\bar{\mathbf{z}}_{<k}$ (with $\bar{\mathbf{z}}_k = F_k(\bar{\mathbf{z}}_{<k})$) yields

$$\mathbf{e}_k \;\coloneqq\; \mathbf{z}_k - \bar{\mathbf{z}}_k \;\approx\; \mathbf{P}_k\Big(J_k\,\mathbf{e}_{<k} + \boldsymbol{\eta}_k\Big), \qquad J_k \;=\; \Big(\frac{\partial g}{\partial \mathbf{z}_{<k}}\Big)_{\bar{\mathbf{z}}_{<k}}, \tag{12}$$

where $\mathbf{e}_{<k}$ stacks the prefix errors and $\boldsymbol{\eta}_k$ collects local modeling/integration error. *Proof.* Chain rule with Lemma 2.

**Corollary (No scale-channel cascade; norm bound).**   Writing $J_k\mathbf{e}_{<k} + \boldsymbol{\eta}_k = \alpha_k\,\bar{\mathbf{z}}_k + \mathbf{t}_k$ with $\mathbf{t}_k \in T_{\bar{\mathbf{z}}_k}\mathbb{S}_R^{d-1}$,

$$\mathbf{e}_k \;\approx\; \mathbf{P}_k(\alpha_k\,\bar{\mathbf{z}}_k + \mathbf{t}_k) \;=\; \mathbf{t}_k, \quad \text{and} \quad \|\mathbf{e}_k\|_2 \;\leq\; \|\mathbf{P}_k J_k\|_2\,\|\mathbf{e}_{<k}\|_2 + \|\mathbf{P}_k\boldsymbol{\eta}_k\|_2. \tag{13}$$

Thus radial (scale) errors are annihilated before refeeding and cannot cascade along the AR chain; only directional (tangential) errors propagate.

**Scope.**   Statements are local (first-order) at points on $\mathbb{S}_R^{d-1}$ and assume $g$ is $C^1$ near $\bar{\mathbf{z}}_{<k}$.

## B   GAUSSIAN POSTERIOR WITH POST-HOC NORMALIZATION: A LOOSER BOUND

We compare the "Gaussian+norm" objective

$$\mathcal{L}_{\mathrm{G}}(\phi, \psi; \mathbf{x}) = \mathbb{E}_{q_\phi(\mathbf{z}|\mathbf{x})}\big[\log p_\psi(\mathbf{x} \mid N_R(\mathbf{z}))\big] - D_{\mathrm{KL}}\big(q_\phi(\mathbf{z} \mid \mathbf{x}) \,\|\, p(\mathbf{z})\big), \qquad N_R(\mathbf{z}) = \frac{R\,\mathbf{z}}{\|\mathbf{z}\|_2},$$

with the spherical ELBO

$$\mathcal{L}_{\mathrm{S\text{-}VAE}}(\phi, \psi; \mathbf{x}) = \mathbb{E}_{q_\phi(\mathbf{u}|\mathbf{x})}\big[\log p_\psi(\mathbf{x} \mid R\mathbf{u})\big] - D_{\mathrm{KL}}\big(q_\phi(\mathbf{u} \mid \mathbf{x}) \,\|\, \mathrm{Unif}(\mathbb{S}^{d-1})\big), \quad \mathbf{u} \in \mathbb{S}^{d-1}.$$

Write the polar decomposition $\mathbf{z} = (r, \mathbf{u})$ with $r = \|\mathbf{z}\|_2 \in \mathbb{R}_{\geq 0}$ and $\mathbf{u} = \mathbf{z}/\|\mathbf{z}\|_2 \in \mathbb{S}^{d-1}$. Let $q_\phi(\mathbf{u} \mid \mathbf{x})$ be the pushforward of $q_\phi(\mathbf{z} \mid \mathbf{x})$ under $\mathbf{z} \mapsto \mathbf{u}$. Since $N_R(\mathbf{z}) = R\mathbf{u}$ depends only on direction, the reconstruction terms coincide:

$$\mathbb{E}_{q_\phi(\mathbf{z}|\mathbf{x})}\big[\log p_\psi(\mathbf{x} \mid N_R(\mathbf{z}))\big] = \mathbb{E}_{q_\phi(\mathbf{u}|\mathbf{x})}\big[\log p_\psi(\mathbf{x} \mid R\mathbf{u})\big].$$

---

[3] For numerical robustness we implement $N_R(\mathbf{z}) = R\,\mathbf{z}/\max(\|\mathbf{z}\|_2, \varepsilon)$ with $\varepsilon = 10^{-7}$.

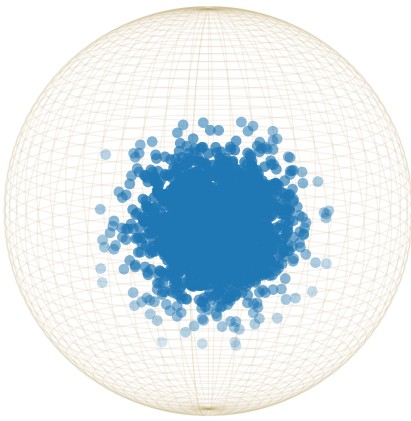
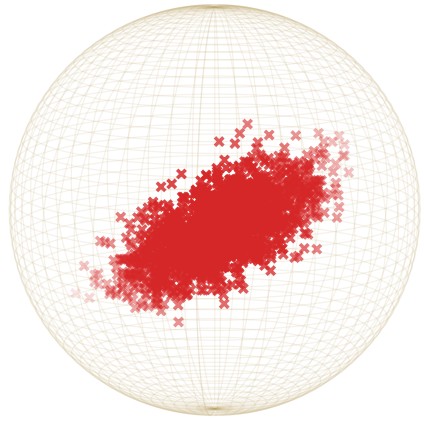

(a) Power Spherical density (axially symmetric about $\boldsymbol{\mu}$).

(b) Gaussian with post-hoc normalization (projected normal / ACG): elliptical level sets, not axially symmetric.

Figure 5: Directional distributions on the sphere. **Left:** Power Spherical respects purely directional geometry—the density depends only on $\boldsymbol{\mu}^\top\mathbf{u}$ and is axially symmetric, with a single concentration parameter $\kappa$. **Right:** Gaussian+norm induces a projected-normal (ACG) law whose level sets follow $\mathbf{u}^\top\boldsymbol{\Sigma}^{-1}\mathbf{u}$; symmetry axes are determined by $\boldsymbol{\Sigma}$ (and the Gaussian mean), so the density is typically elliptical rather than axially symmetric.

For the isotropic Gaussian prior $p(\mathbf{z}) = \mathcal{N}(\mathbf{0}, \mathbf{I})$, one has $p(\mathbf{z}) = p(r)\,\mathrm{Unif}(\mathbf{u})$ (with $p(r)$ the $\chi$-law in $\mathbb{R}^d$). The KL chain rule (disintegration over $\mathbf{u}$) gives

$$D_{\mathrm{KL}}\big(q_\phi(\mathbf{z} \mid \mathbf{x}) \,\|\, p(\mathbf{z})\big) = D_{\mathrm{KL}}\big(q_\phi(\mathbf{u} \mid \mathbf{x}) \,\|\, \mathrm{Unif}(\mathbb{S}^{d-1})\big) + \mathbb{E}_{q_\phi(\mathbf{u}\mid\mathbf{x})}\Big[D_{\mathrm{KL}}\big(q_\phi(r \mid \mathbf{u}, \mathbf{x}) \,\|\, p(r)\big)\Big].$$

Combining the two displays yields

$$\mathcal{L}_{\mathrm{G}}(\phi, \psi; \mathbf{x}) = \mathcal{L}_{\mathrm{S\text{-}VAE}}(\phi, \psi; \mathbf{x}) - \mathbb{E}_{q_\phi(\mathbf{u}\mid\mathbf{x})}\Big[D_{\mathrm{KL}}\big(q_\phi(r \mid \mathbf{u}, \mathbf{x}) \,\|\, p(r)\big)\Big] \;\leq\; \mathcal{L}_{\mathrm{S\text{-}VAE}}(\phi, \psi; \mathbf{x}),$$

with equality only if $q_\phi(r \mid \mathbf{u}, \mathbf{x}) = p(r)$ almost surely (i.e., the posterior's radial law exactly matches the prior and is independent of $\mathbf{x}, \mathbf{u}$). Thus, Gaussian posterior with post-hoc normalization pays an extra nonnegative *radial* KL penalty while the decoder discards radius; a spherical posterior avoids this mismatch and aligns the bound with the constant-norm constraint.

**Remark (axial symmetry vs. projected normal on $\mathbb{S}^{d-1}$).** The Power Spherical (PS) density on the unit sphere has the form $f_{\mathrm{PS}}(\mathbf{u}) = Z_d(\kappa)\big(1 + \boldsymbol{\mu}^\top\mathbf{u}\big)^\kappa$ with $\|\boldsymbol{\mu}\|_2 = 1$ and $\kappa \geq 0$. It is *axially rotationally symmetric* about $\boldsymbol{\mu}$: for any rotation $Q$ with $Q\boldsymbol{\mu} = \boldsymbol{\mu}$, one has $f_{\mathrm{PS}}(Q\mathbf{u}) = f_{\mathrm{PS}}(\mathbf{u})$. The single scalar $\kappa$ monotonically controls geodesic concentration (with $\kappa = 0$ yielding the uniform law).

By contrast, *Gaussian+norm*—take $\mathbf{z} \sim \mathcal{N}(\boldsymbol{\mu}_g, \boldsymbol{\Sigma})$ in $\mathbb{R}^d$ and map to the sphere via $\mathbf{u} = \mathbf{z}/\|\mathbf{z}\|_2$—induces a projected-normal (Angular Central Gaussian, ACG) directional law. For the zero-mean case ($\boldsymbol{\mu}_g = \mathbf{0}$), its density is $f_{\mathrm{ACG}}(\mathbf{u}) \propto (\mathbf{u}^\top\boldsymbol{\Sigma}^{-1}\mathbf{u})^{-d/2}$: level sets follow the quadratic form $\mathbf{u}^\top\boldsymbol{\Sigma}^{-1}\mathbf{u}$ and are generally *elliptical*, not axially symmetric; axial symmetry holds only if $\boldsymbol{\Sigma} \propto \mathbf{I}$ (then the law is uniform). With nonzero mean $\boldsymbol{\mu}_g \neq \mathbf{0}$, the density also depends on $\boldsymbol{\mu}_g^\top\boldsymbol{\Sigma}^{-1}\mathbf{u}$, producing skewed, non-axially symmetric level sets (and, for anisotropic $\boldsymbol{\Sigma}$, possible antipodal bimodality when $\boldsymbol{\mu}_g = \mathbf{0}$). Therefore the Power Spherical family matches the intended purely directional geometry on $\mathbb{S}^{d-1}$, whereas Gaussian+norm inherits Euclidean anisotropy from $(\boldsymbol{\mu}_g, \boldsymbol{\Sigma})$ and does not.

## C   COMPARISON WITH MAR'S VAE

We perform an ablation that swaps our S-VAE for MAR's VAE (Li et al., 2024a) while keeping the AR backbone, training schedule, and evaluation protocol identical (see Sec. 4.3). As summarized in Table 3, S-VAE yields a large FID gain ($4.54 \rightarrow 2.52$) and higher IS ($241.6 \rightarrow 258.4$). These results indicate that constant-norm, directional latents from S-VAE materially strengthen continuous-token AR generation.

Table 3: Ablation: swapping our S-VAE for MAR's VAE *without retraining* (same AR backbone and training/evaluation settings; only the VAE changes).

| Type | FID↓ | IS↑ | Pre.↑ | Rec.↑ |
|------|------|-----|-------|-------|
| AR + MAR's VAE | 4.54 | 241.6 | 0.84 | 0.45 |
| **AR + S-VAE (ours)** | **2.52** | **258.4** | 0.82 | 0.56 |

Table 4: Comparison of VAE backbones. Training speed measured in iterations per second on 8 A100 GPUs, with batch size 256.

| Type | #Params | Training Speed↑ | LPIPS↓ | FID↓ | IS↑ | Pre.↑ | Rec.↑ |
|------|---------|-----------------|--------|------|-----|-------|-------|
| CNN | 70M | 2.25 | 0.167 | 2.63 | **262.6** | 0.82 | 0.55 |
| ViT | 170M | **7.19** | 0.178 | 2.81 | 251.3 | 0.82 | 0.55 |
| **Hybrid (ours)** | 75M | 5.81 | **0.166** | **2.52** | 258.4 | 0.82 | 0.56 |

## D  VAE ARCHITECTURE — CNN VS. VIT VS. HYBRID

Most latent VAEs for image generation adopt a VQGAN-style encoder-decoder (Esser et al., 2021), i.e., a convolutional (CNN) backbone with downsampling/upsampling blocks. This design is parameter-efficient ($\sim$70M) but throughput is often limited by activation memory and bandwidth on large feature maps, leading to slow training and inference. ViT-VQGAN (Yu et al., 2021) replaces the CNN backbone with a Vision Transformer (ViT) to improve efficiency; however, as shown in Table 4, a pure ViT backbone yields weaker generative metrics than a CNN.

To balance efficiency and quality, we adopt a *hybrid* VAE architecture. The encoder first uses a lightweight CNN stem (with downsampling blocks) for *patchification* and early spatial mixing, imparting CNN inductive bias while reducing spatial resolution. The resulting patch tokens are then processed by a bidirectional Transformer (ViT blocks) to model long-range dependencies. The decoder mirrors this design: a ViT stack refines the latent tokens, followed by a lightweight CNN (with upsampling blocks) for *unpatchification* and pixel-level reconstruction. This hybrid preserves the CNN's strong local bias while leveraging the ViT's global receptive field at token resolution, yielding a favorable speed-quality tradeoff.

In our implementation, to match the parameter scale of a VQGAN-style CNN encoder-decoder (Esser et al., 2021), our S-VAE uses 6 ViT blocks in the encoder and 12 in the decoder, each with hidden size 512. The encoder's CNN stem performs patchification via 4 downsampling stages (overall $16\times$ reduction) with channel widths $[64, 64, 128, 256, 512]$; the decoder mirrors this with 4 upsampling stages and an extra residual block per stage. In total, the S-VAE has $\sim$75M parameters.

We compare three encoder-decoder backbones under the same training setup and losses: (i) a VQGAN-style CNN (Esser et al., 2021); (ii) a pure ViT (Yu et al., 2021) (12 Transformer blocks in both encoder and decoder, hidden size 768; $\sim$170M params); and (iii) our *Hybrid* design. To reduce compute, the training follows the setting in Sec. 4.3. We report training throughput (iterations/s) of VAE, perceptual distortion (LPIPS with VGG-16), and downstream ImageNet generative metrics. Results in Table 4 show: *ViT* is fastest (7.19 it/s) but slightly worse on LPIPS/FID/IS; the *CNN* is slowest (2.25 it/s) yet competitive in IS; our *Hybrid* retains the best reconstruction (LPIPS 0.166) and the best FID (2.52) while running at 5.81 it/s—about $2.6\times$ faster than the CNN and at $81\%$ of ViT throughput (5.81 vs 7.19 it/s). Overall, the hybrid backbone offers the most favorable speed-quality trade-off.

## E  IMAGENET $512\times512$

Following prior work, we also evaluate on ImageNet at $512 \times 512$ resolution. We train S-VAE at $512 \times 512$ with hidden dimension $d = 16$ and a KL coefficient $\beta = 0.016$. For the autoregressive transformer, we adopt the *SphereAR-L* configuration and set the number of class-conditional tokens to 64. Unless otherwise noted, all other settings mirror the $256 \times 256$ experiments. As shown in Table 5, *SphereAR-L* achieves an FID of 1.67 competitive with prior models.

Table 5: Overall comparison on ImageNet 512×512 class-conditional generation.

| Model | Type | Order | #Params | #Epochs | FID↓ | IS↑ |
|---|---|---|---|---|---|---|
| *Discrete Tokens* | | | | | | |
| MaskGIT (Chang et al., 2022) | Mask. | random | 227M | 300 | 4.46 | **342.0** |
| MAGVIT-v2 (Yu et al., 2023) | Mask. | random | 307M | 270 | 1.91 | 324.3 |
| VAR-d36-s (Tian et al., 2024) | N.S. | - | 2.3B | 350 | 2.62 | 303.2 |
| *Continuous Tokens* | | | | | | |
| DiT-XL/2 (Peebles and Xie, 2023) | Diff. | - | 675M | 400 | 3.04 | 240.8 |
| MAR-L (Li et al., 2024a) | Mask. | random | 481M | 800 | 1.73 | 279.9 |
| *SphereAR-L* (Our) | AR | raster | 481M | 400 | **1.67** | 292.5 |

# F  TRAINING CONVERGENCE EFFICIENCY

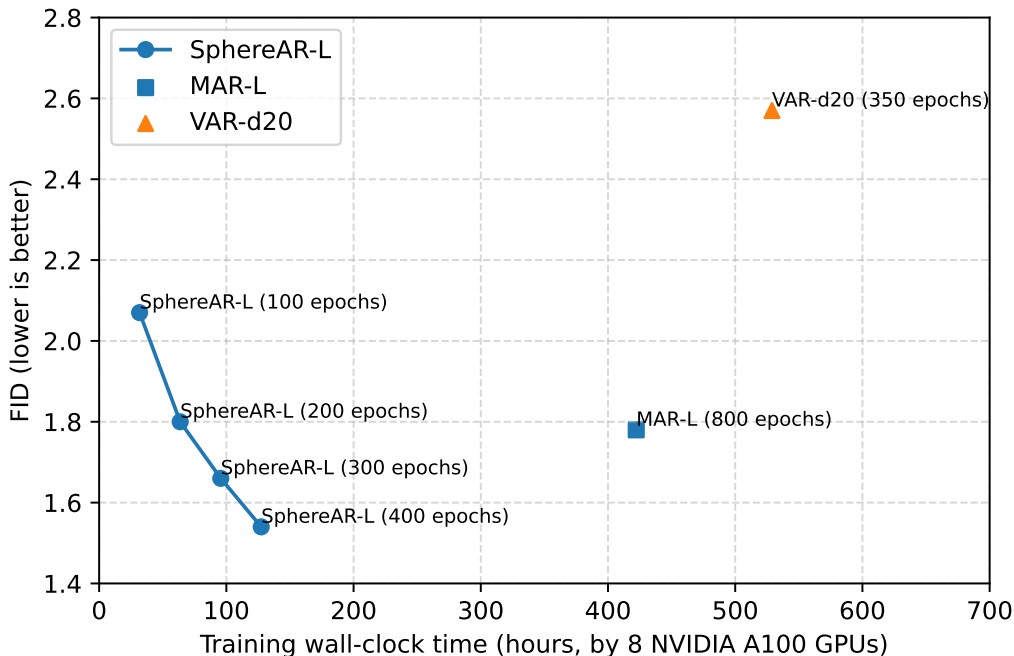

Figure 6: Training Convergence Efficiency Comparison. We plot the generative performance (FID, lower is better) as a function of training cost (wall-clock time) for SphereAR (our model), MAR, and VAR. SphereAR converges to a strong FID score significantly faster than both baselines.

One key advantage of autoregressive (AR) models is their high training efficiency. More specifically, for a single training sample, an AR model calculates a loss across all positions of the sequence (i.e., all prediction steps) in one forward pass (using teacher forcing). In contrast, diffusion or masked-based models merely compute a loss for a single sampled step (e.g., one timestep t in diffusion or one mask pattern). Therefore, because diffusion or masked-based models receive a less dense supervisory signal per sample, they often require significantly more epochs to converge.

To explicitly demonstrate this, we additionally evaluated the generative performance of intermediate checkpoints during the training of SphereAR. For a fair comparison, all wall-clock training times were measured using 8 NVIDIA A100 GPUs. As shown in Fig. 6, SphereAR converges much faster than MAR and VAR. This efficiency is stark: in terms of epochs, SphereAR (200 epochs) achieves comparable performance to MAR-L (800 epochs). In terms of wall-clock time, SphereAR achieves this same performance using only ∼20% of the training cost required by MAR-L.

# G  INFERENCE SPEED

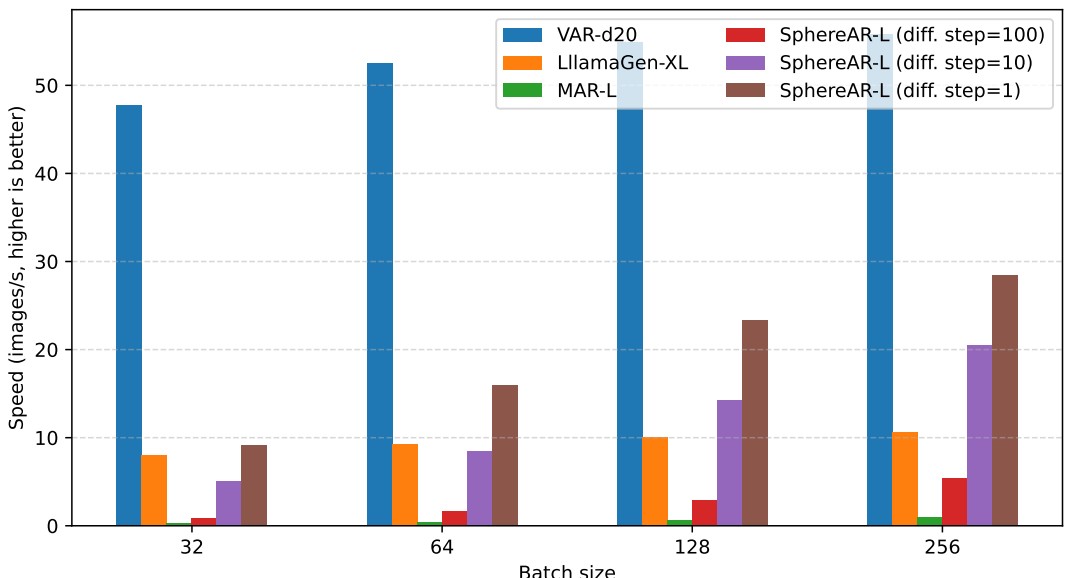

Figure 7: Inference speed comparison on a single A100 GPU. SphereAR's speed, which depends on the number of diffusion steps, is benchmarked against MAR (slowest), VAR (fastest), and LlamaGen.

This section benchmarks the inference speed of our model against previous approaches. All benchmarks were performed on a single NVIDIA A100 GPU across various batch sizes. The results, presented in Fig. 7, lead to the following observations: (i) VAR is the fastest model, as it only requires dozens of steps to generate an image. The purely autoregressive model LlamaGen is also highly efficient. (ii) MAR is the slowest. This is expected, as its architecture requires accessing the full-length token sequence at each sampling step, which is compounded by an additional 100 sampling steps in its diffusion head. (iii) For SphereAR, our default configuration (with 100 sampling steps in diffusion head) is significantly faster than MAR but slower than LlamaGen due to the computational overhead of the diffusion head. (iv) We also evaluated two additional SphereAR settings (with 10 and 1 diffusion steps, respectively), which exhibit progressively faster speeds. This result clearly demonstrates that the primary inference bottleneck lies in the diffusion head.

These observations highlight several paths for optimization. First, the high sampling cost of the diffusion head can be mitigated by integrating recent few-step diffusion models, such as Shortcut-Model (Frans et al., 2024) or MeanFlow (Geng et al., 2025), which dramatically reduce the required number of steps. Second, since the core of our model is a standard autoregressive Transformer (akin to other LLMs), it can directly benefit from established inference optimization libraries (e.g., vLLM) to further accelerate the token generation process.

# H  ROBUSTNESS WITH HIGH CFG SCALES

To demonstrate SphereAR's robustness to high CFG scales, we conducted an additional experiment. In particular, we ran sampling with progressively higher CFG scales for the best diagonal-Gaussian model (G-16) and our best S-VAE model (S-04) from Sec. 4.3. The results are presented in Fig. 8.

As shown in the figure, the performance of both models' degrades as the CFG scale increases. However, the rate of degradation for S-04 is much slower. This is evidenced by the widening performance gap: at a CFG scale of 4.0, the gap between the models is ~1.0, but at a CFG scale of 8.0, it widens to ~2.0. This result clearly demonstrates that the proposed model is more robust to aggressive CFG scales.

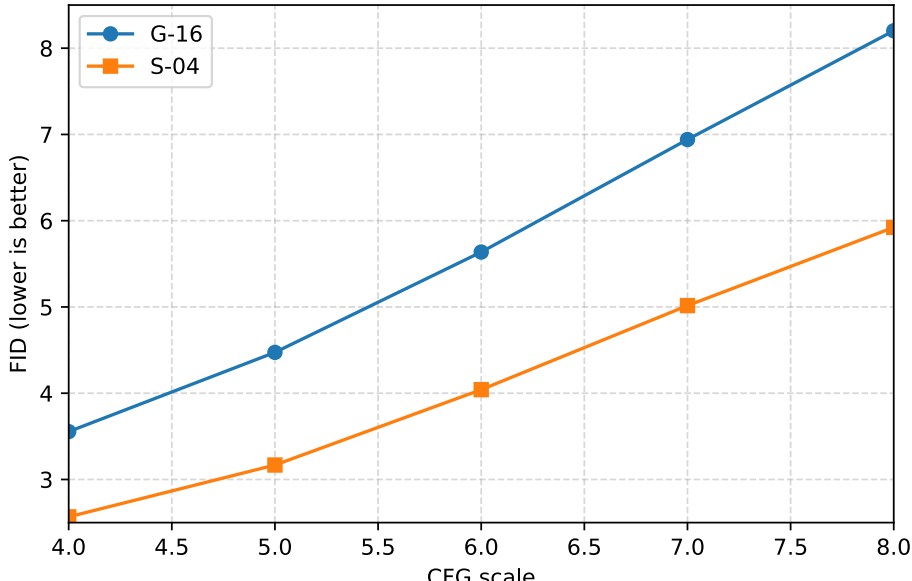

Figure 8: Performance (FID, lower is better) versus CFG scale, comparing our S-VAE model ('S-04') with the Gaussian baseline ('G-16'). The curves demonstrate that 'S-04' is more robust to high CFG values.

## I   FAILURE CASES AND LIMITATIONS

We present a qualitative analysis of failure modes in Fig. 9. Similar to existing methods, SphereAR can produce results with noticeable artifacts, especially in complex classes. However, even in these failure cases, SphereAR-L consistently produces visually superior results with fewer artifacts than the baseline MAR-L.

Despite this relative improvement, both models share common limitations. These primarily include artifacts in human faces and, most prominently, a failure to render legible text or other high-frequency details (e.g., street signs, bottle labels), which are often generated as semantically meaningless patterns.

We posit this is largely a data-driven limitation. As noted by MAR (Li et al., 2024a), models trained on academic datasets like ImageNet often show a "noticeable gap in visual quality" compared to commercial models trained on massive, diverse web data, especially for categories like coherent text and faces. These failures are thus likely attributable to the dataset's limitations rather than a fundamental flaw in our architecture.

Due to computational constraints, our current evaluation is focused on the ImageNet benchmark. Therefore, assessing the scalability of SphereAR and its robustness in more diverse, large-scale data regimes remains a critical direction for future investigation.

## J   USE OF LLMS

We used a large language model (OpenAI's ChatGPT) solely for language polishing—proofreading, minor rephrasing, and tone consistency. It did not generate or influence the scientific ideas, methods, results, or conclusions. All suggestions were reviewed and edited by the authors, who take full responsibility for the paper's content and integrity.

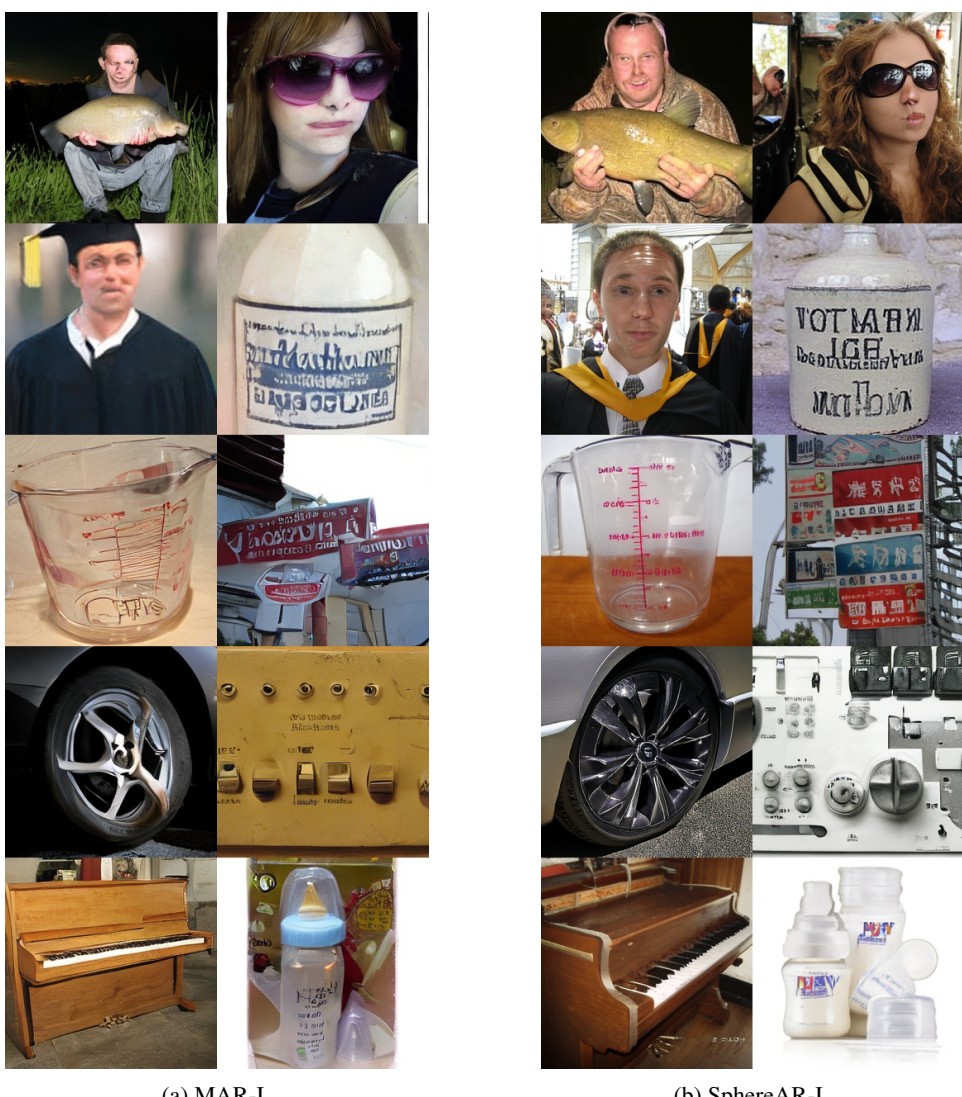

(a) MAR-L                    (b) SphereAR-L

Figure 9: **Failure cases**. Similar to existing methods, our model can produce results with noticeable artifacts. We show MAR-L (left) and SphereAR-L (right) 's results of the same classes.

## K    MODEL GENERATED EXAMPLES

We show the uncurated 256×256 samples generated by our 479M *SphereAR-L*, from Fig. 10 to Fig. 21.

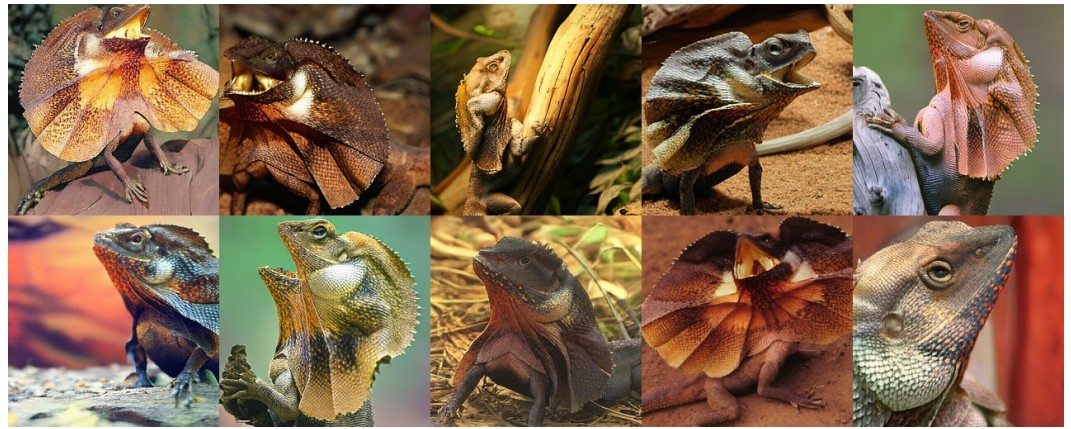

Figure 10: Uncurated 256×256 *SphereAR-L* samples. Class label: "frilled lizard" (43).

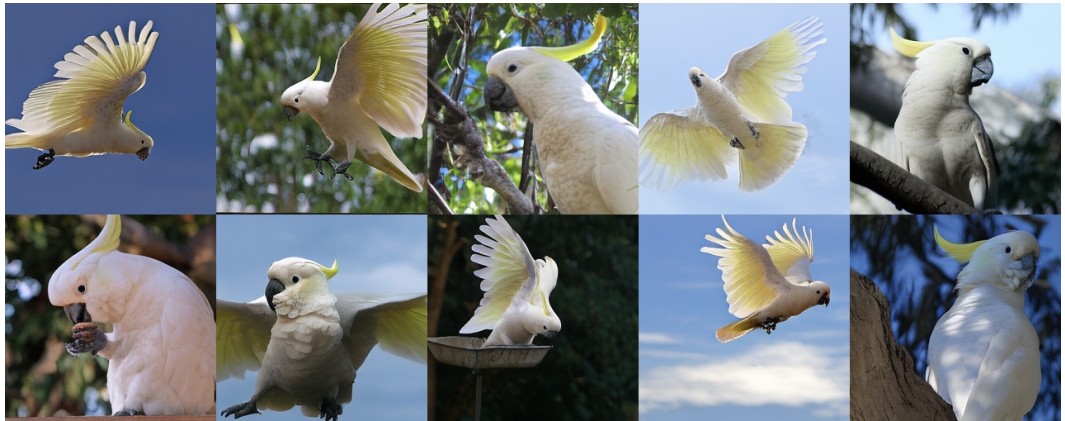

Figure 11: Uncurated 256×256 *SphereAR-L* samples. Class label: "sulphur-crested cockatoo" (89).

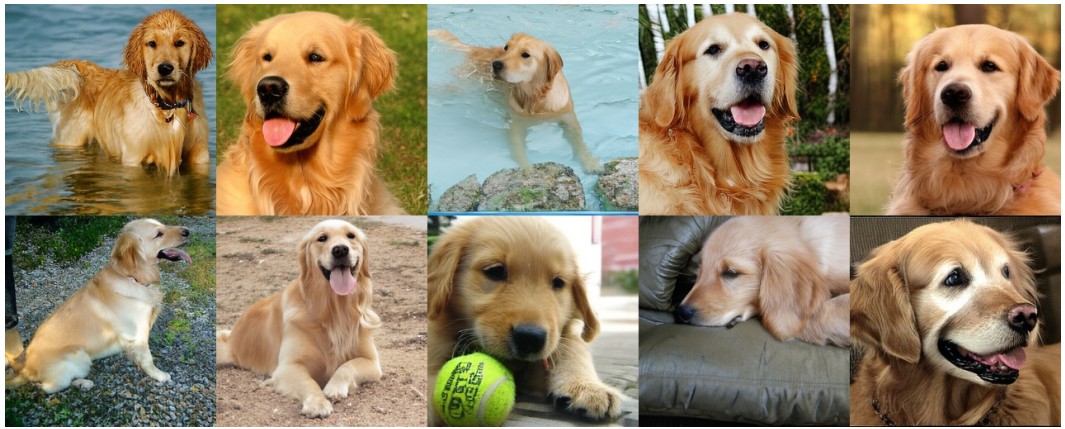

Figure 12: Uncurated 256×256 *SphereAR-L* samples. Class label: "golden retriever" (207).

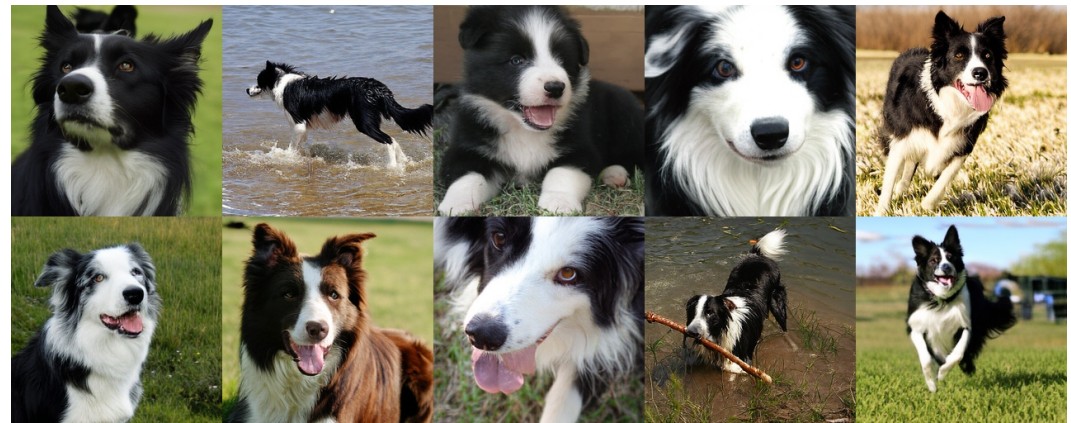

Figure 13: Uncurated 256×256 *SphereAR-L* samples. Class label: "Border collie" (232).

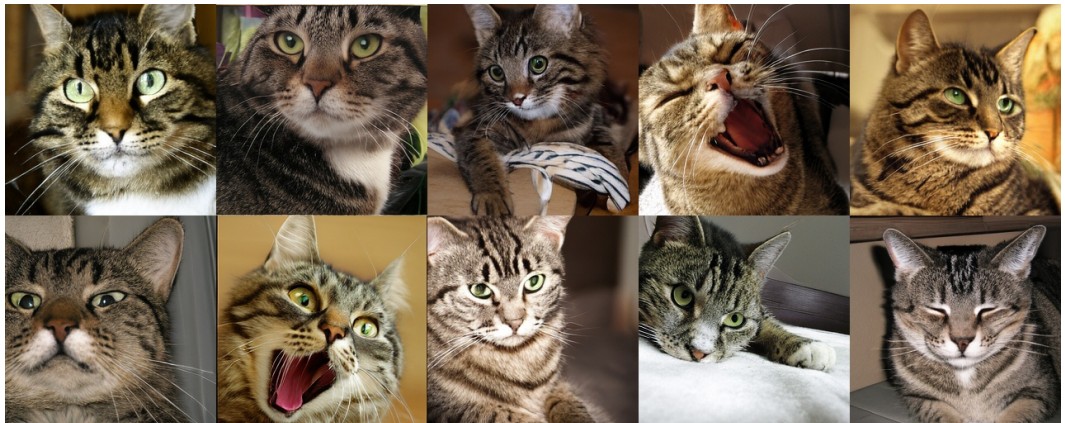

Figure 14: Uncurated 256×256 *SphereAR-L* samples. Class label: "tabby cat" (281).

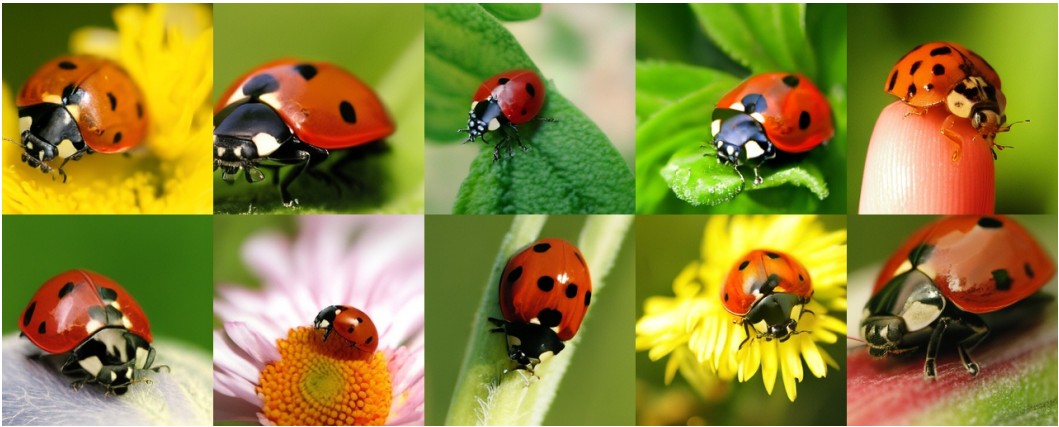

Figure 15: Uncurated 256×256 *SphereAR-L* samples. Class label: "ladybug" (301).

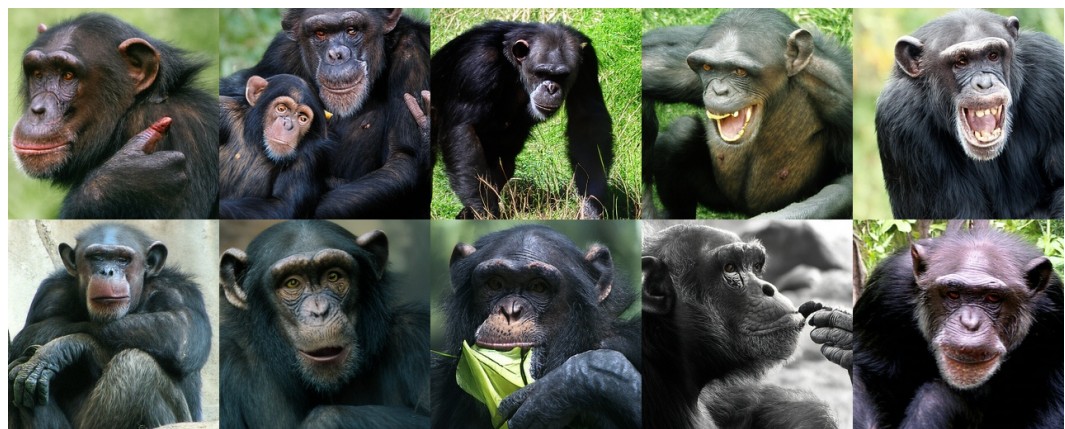

Figure 16: Uncurated 256×256 *SphereAR-L* samples. Class label: "chimpanzee" (367).

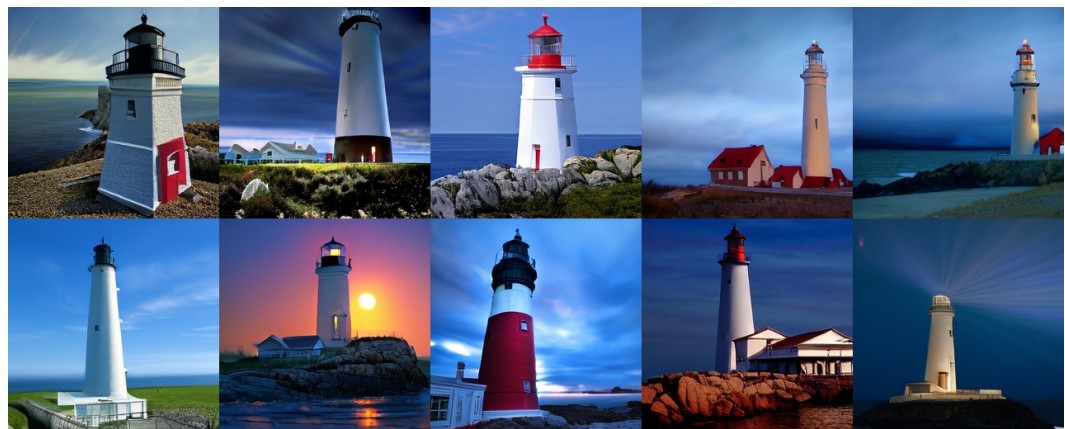

Figure 17: Uncurated 256×256 *SphereAR-L* samples. Class label: "beacon" (437).

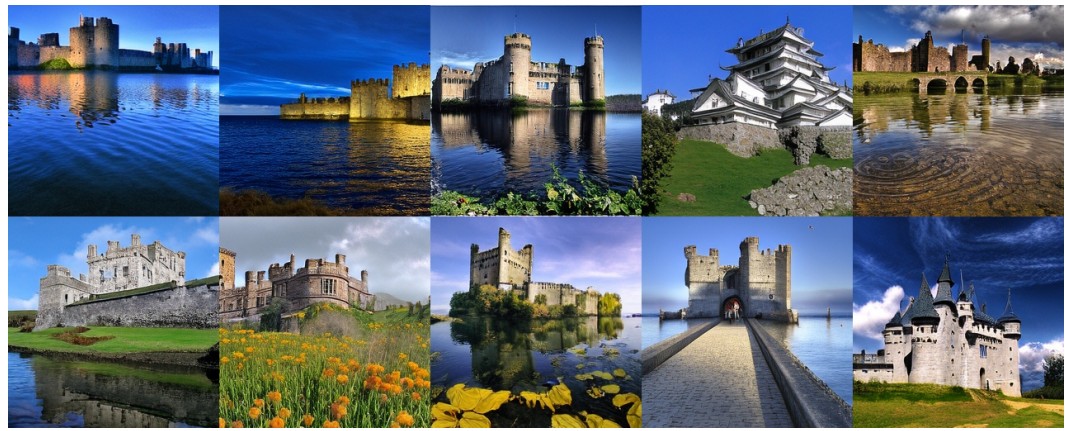

Figure 18: Uncurated 256×256 *SphereAR-L* samples. Class label: "castle" (483).

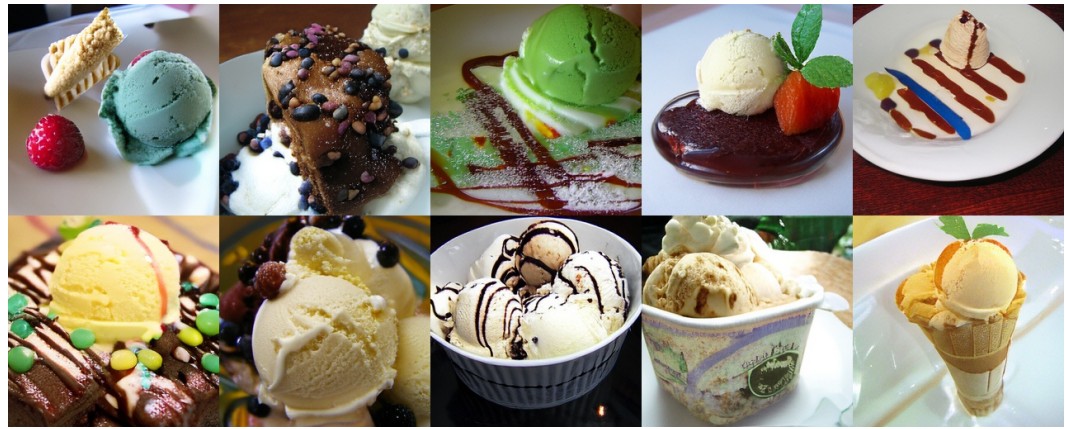

Figure 19: Uncurated 256×256 *SphereAR-L* samples. Class label: "icecream" (928).

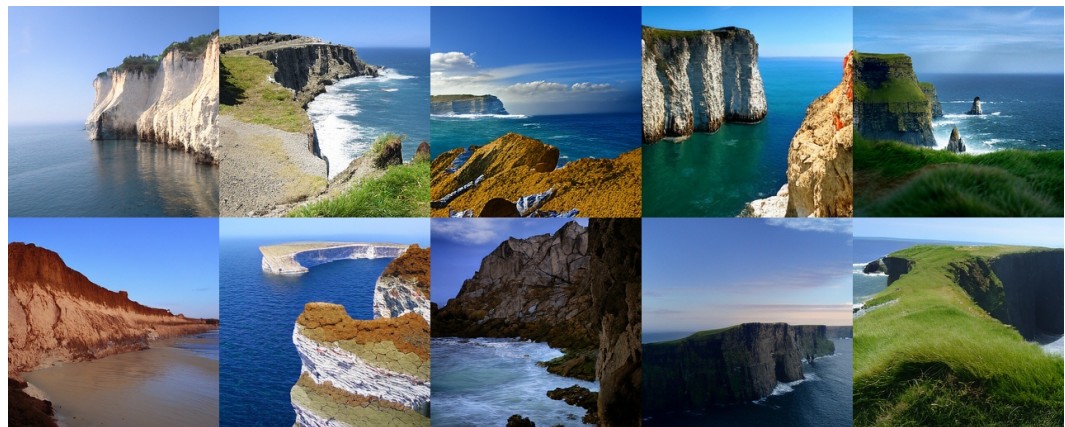

Figure 20: Uncurated 256×256 *SphereAR-L* samples. Class label: "cliff" (972).

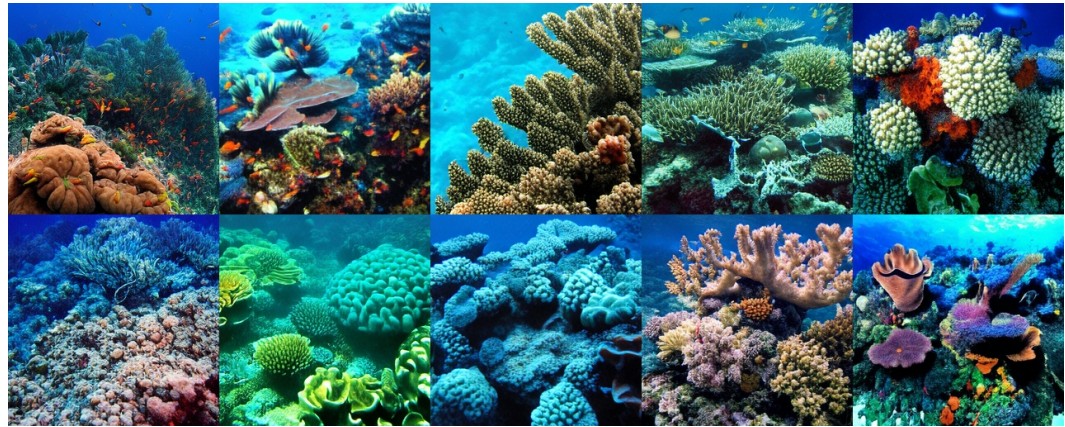

Figure 21: Uncurated 256×256 *SphereAR-L* samples. Class label: "coral reef" (973).

