# OpenReview forum: "Hyperspherical Latents Improve Continuous-Token Autoregressive Generation"
_ICLR.cc/2026/Conference — ICLR 2026 Poster_

### Official Review · Reviewer_LHMS · 2025-10-22

**Soundness:** 4
**Presentation:** 4
**Contribution:** 4
**Rating:** 8
**Confidence:** 4

**Summary:**

The paper proposes SphereAR, a continuous-token autoregressive image generator that prevents variance collapse by enforcing scale-invariant latents: a hyperspherical VAE constrains each token to a fixed-radius hypersphere, and all AR inputs/outputs—including after classifier-free guidance—are projected back to this sphere. A first-order analysis shows normalization removes radial errors, stabilizing multi-step AR decoding. Coupled with a causal Transformer and a token-level diffusion head, SphereAR sets impressive results on ImageNet 256×256: SphereAR-L achieves FID 1.54, surpassing diffusion and masked-generation baselines and matching MAR-H with ~1/2 the parameters. Ablations confirm hyperspherical latents outperform diagonal-Gaussian, fixed-variance, and Gaussian+post-hoc normalization, and that constant-norm refeeding in AR is the key driver of gains.

**Strengths:**

* This paper introduces a simple but powerful principle—enforcing constant-norm latents for continuous-token AR, including after CFG—to remove scale degrees of freedom. It provides a clean theoretical first-order analysis explaining why normalization eliminates radial error accumulation. Besides, it also distinguishes S-VAE from Gaussian+post-hoc normalization and shows why the former is better aligned with the objective.

* Strong empirical evidence on ImageNet 256×256 with clear SOTA FID at comparable or fewer parameters; comprehensive ablations that validate the effectiveness of each proposed component.

* Well-motivated problem statement tied to variance collapse under CFG; intuitive figures and precise method description; appendices with proofs and architectural details; explicit training and inference settings with reproducibility plan.

**Weaknesses:**

I think this paper does not contain significant weaknesses. I have a few suggestions and questions:
* It would be better to provide wall-clock (efficiency) comparisons vs MAR/VAR/DiT at matched FID.
* It would be interesting to see if the proposed method can improve the MAE paradigm as well.

**Questions:**

Please refer to the above section.

---

> ### Author Response · Authors · 2025-11-20
>
> We sincerely thank the reviewer for the support of our work. We address your specific suggestions below:
>
> > **Q1: "It would be better to provide wall-clock (efficiency) comparisons vs MAR/VAR/DiT at matched FID."**
>
> **Response:**
> We thank the reviewer for this constructive suggestion. We have added a detailed wall-clock efficiency analysis in **Appendix F**.
>
> - **Training Efficiency:** Our experiments demonstrate that SphereAR-L is highly efficient: it achieves the same FID performance as MAR-L with **only $\sim$20% of the training cost**.
> - **Inference Efficiency:** We have also provided a detailed comparison of inference speeds against MAR, VAR, and LlamaGen in **Appendix G**. The results show that SphereAR significantly outperforms MAR and approaches the throughput of other baselines when using optimized settings.
>
> > **Q2: "It would be interesting to see if the proposed method can improve the MAE paradigm as well."**
>
> **Response:**
> **We interpret "MAE" as a typo for "MAR,"** since standard MAE is primarily a representation learning framework.
>
> We believe **S-VAE could indeed improve MAR**. Since MAR relies on iterative generation, it is also susceptible to **error accumulation** caused by CFG rescaling, albeit less pronounced than in standard AR due to fewer steps. Therefore, the geometric alignment provided by S-VAE should theoretically offer stability benefits.
>
> However, due to the high computational cost of retraining MAR, this experiment exceeds the feasible scope of the rebuttal period. We leave this validation for future work.

---

> > ### Comment · Reviewer_LHMS · 2025-11-22
> >
> > Thanks for your response. Yes I do mean MAR instead of MAE. I will keep my positive score.

---

### Official Review · Reviewer_xjkB · 2025-11-01

**Soundness:** 4
**Presentation:** 4
**Contribution:** 3
**Rating:** 6
**Confidence:** 3

**Summary:**

This paper tackles a core limitation in continuous-token autoregressive (AR) image generation—variance collapse caused by heterogeneous latent scales during AR decoding, especially under classifier-free guidance. The authors propose **SphereAR**, which enforces a constant-norm hyperspherical latent space via a hyperspherical VAE (S-VAE) and ensures all AR inputs and outputs are projected back onto a fixed-radius hypersphere. The paper provides theoretical justification showing that normalization eliminates radial error propagation in AR decoding and supports it with extensive empirical evidence on ImageNet-256×256. SphereAR-L (479M) achieves FID 1.54, surpassing diffusion and masked-generation baselines of comparable or even larger parameter scales (e.g., MAR-H, VAR-d30, DiT-XL/2). Ablations further show that hyperspherical latent geometry consistently improves stability and generation quality compared to Gaussian posteriors or post-hoc normalization.

**Strengths:**

* **Clear motivation with strong theoretical insight.** The paper identifies *scale drift* as the fundamental issue in continuous-token AR and justifies why hyperspherical constraint solves it more elegantly than variance inflation or post-hoc normalization.
* **Significant empirical improvement.** SphereAR sets a new state-of-the-art among AR models and even outperforms strong diffusion and masked-generation baselines with fewer parameters.
* **Well-structured ablation study.** Experiments isolate the contribution of (i) spherical posterior vs Gaussian, (ii) normalization applied only at VAE decoder input vs AR loop, and (iii) post-hoc normalization vs intrinsic spherical latent modeling.
* **Technical contribution beyond engineering tweak.** Introducing hyperspherical VAEs into AR pipelines constitutes a genuinely novel design in continuous-token generative modeling.
* **Good reproducibility.** Training details, architectural configurations, and evaluation settings are consistently documented.

**Weaknesses:**

* **Limited analysis beyond ImageNet.** All experiments are class-conditional ImageNet generation at 256×256. It is unclear whether SphereAR scales to higher-resolution tasks or more challenging datasets such as LAION-based text-to-image.
* **AR decoding speed not fully addressed.** While normalization improves stability, AR inference remains autoregressive with diffusion steps per token, leading to slower sampling compared to parallel masked generation or next-scale diffusion models. A quantitative runtime comparison would be helpful.
* **Lack of negative results on extreme CFG regimes or resolution scaling.** The theoretical argument suggests strong stability, but empirical evaluation focuses on standard CFG ranges.
* **S-VAE training complexity.** Hyperspherical posteriors (even using Power Spherical) may increase training difficulty relative to standard diagonal-Gaussian VAEs; the paper acknowledges this but does not quantify the stability or convergence trade-off.

**Questions:**

Please refer to the "Weakness".

---

> ### Author Response · Authors · 2025-11-20
>
> We sincerely thank the reviewer for the positive assessment for our submission.
> We address your specific concerns below:
>
> > **Q1: "scales to higher-resolution tasks or more challenging datasets."**
>
> **Response:**
> We thank the reviewer for this valuable suggestion regarding scalability. To demonstrate the robustness of SphereAR, we have conducted **two additional experiments**:
>
> - **(a) Scaling to Higher Resolution:** As shown in **Appendix E**, we extended SphereAR to ImageNet $512 \times 512$ generation. It achieves an **FID of 1.67**, outperforming previous baselines.
> - **(b) Scaling Model Size:** To compare fairly with MAR-H (943M), we trained a larger model, **SphereAR-H (943M)**. As reported in Table 1, SphereAR-H achieves an **FID of 1.34**, setting a new SOTA for autoregressive generation.
>
> These results strongly suggest that SphereAR scales effectively with both resolution and model capacity.
>
> **Regarding text-to-image generation (e.g., on LAION),** we agree this is an important direction. It is worth noting that SphereAR employs a **standard autoregressive Transformer** architecture, making it inherently compatible with text-to-image tasks. Extending it to handle text prompts is structurally straightforward, similar to established methods like LlamaGen.
>
> However, given that ImageNet is the standard benchmark for validating image generation architectures (e.g., DiT, VAR, MAR), and due to the immense computational resources required for large-scale text-to-image training, we leave this for future work. We will release our code and model weights to enable the community to explore SphereAR on these diverse tasks.
>
>
> > **Q2: “AR decoding speed not fully addressed... A quantitative runtime comparison would be helpful.”**
>
> **Response:**
> We thank the reviewer for this suggestion. We have included a detailed quantitative runtime comparison in **Appendix G**.
>
> Our results offer a nuanced view compared to the general intuition that "AR is slower than masked generation":
>
> - **VAR** is indeed the fastest due to its next-scale prediction nature.
> - **LlamaGen** (standard discrete AR) is also highly efficient.
> - **MAR** (Masked AR), surprisingly, was found to be the **slowest** in our benchmark. This is likely due to the computational overhead of its diffusion head combined with the need to process the full-length token sequence at each iterative step.
>
> **Regarding SphereAR:**
> Our default configuration (100 diffusion steps) is **much faster than MAR**, though slower than LlamaGen due to the diffusion head overhead. However, we demonstrate that this is not a fundamental bottleneck. We evaluated SphereAR with **10 and 1 diffusion steps**, showing progressively faster speeds that approach standard AR models. This can be achieved via recent few-step diffusion models like **Shortcut Models** [1] or **MeanFlow** [2].
>
> [1] Frans et al., "One Step Diffusion via Shortcut Models", 2024.
>
> [2] Geng et al., "Mean Flows for One-step Generative Modeling", 2025.
>
> > **Q3: “Lack of negative results on extreme CFG regimes.”**
>
> **Response:**
> We thank the reviewer for this valuable suggestion. To address this, we conducted an additional experiment examining generative performance under high CFG scales, which is presented in **Appendix H**.
>
> As shown in the results, the performance of both models naturally degrades as the CFG scale increases. However, the **rate of degradation** for SphereAR is significantly slower than the Gaussian-based baseline. This is evidenced by the widening performance gap: at a CFG scale of 4.0, the gap between the models is $\sim$1.0, whereas at a CFG scale of 8.0, it widens to $\sim$2.0.
>
> This result clearly demonstrates that the proposed model is significantly more robust to aggressive CFG values.
>
>
> > **Q4: “S-VAE training complexity... quantify the stability or convergence trade-off.”**
>
> **Response:**
> We thank the reviewer for raising this important point regarding potential trade-offs.
>
> In our experiments, we **did not observe any increase in training instability or computational cost**. We quantify this in two aspects:
>
> 1.  **Computational Efficiency:** The implementation of the Power Spherical distribution is highly efficient. We found that the per-iteration training time of S-VAE is virtually identical to that of the Gaussian-based VAE, indicating negligible computational overhead.
>
> 2.  **Training Stability:** S-VAE serves as a robust "drop-in" replacement. We used **identical hyperparameter settings** (e.g., learning rate, Adam betas, gradient clipping) as the standard Gaussian VAE. Under these conditions, S-VAE converged smoothly without requiring any specific tuning or stabilization tricks.
>
> In short, S-VAE introduces structural benefits without imposing a penalty on training efficiency or stability.

---

### Official Review · Reviewer_Li3p · 2025-11-01

**Soundness:** 4
**Presentation:** 3
**Contribution:** 3
**Rating:** 4
**Confidence:** 3

**Summary:**

This paper presents SphereAR, an autoregressive image generation framework designed to address the instability and performance gap of continuous-token AR models compared to diffusion and masked-generation methods. The key innovation is the use of hyperspherical variational autoencoders (S-VAE) to produce constant-norm latent tokens, enforced throughout AR decoding by projecting all inputs and outputs onto a fixed-radius hypersphere. Theoretical analysis links the scale-invariance to improved stability, and empirical results on ImageNet-1K demonstrate SphereAR’s superior FID and parameter efficiency versus strong baselines. The paper also includes extensive ablations and analyses of architectural choices.

**Strengths:**

1. Theoretical motivation is clear, with the scale-invariance argument spelled out through formal analysis.
2. Strong Empirical Results:  The clear advantage in both FID and parameter efficiency is validated across multiple regimes (base and large model).
3. Comprehensive Experimental Methodology: Empirical claims are supported by well-documented implementation details (Sec. 4.1 & Appendix C) and a reproducibility statement promising code and hyperparameter release.

**Weaknesses:**

1. Limited Discussion of Generalization and Failure Cases: The paper focuses almost exclusively on ImageNet-1K at $256 \times 256$ resolution (see Section 4 and qualitative figures in Appendix E). There is no analysis of generalization to other domains (e.g., other image datasets, non-class-conditional tasks, or higher resolutions), nor qualitative exploration of failure modes. This raises questions about the robustness and transferability of the claimed benefits.
2. Empirical Focus Narrowed to FID and Class-Conditional Generation: Table 1 and Figure 4 concentrate on FID, IS, Precision, and Recall. FID is known to have limitations (especially in capturing semantic consistency and diversity)
3. Ablation Duration and Robustness: The ablation experiments in Section 4.3 are run for a much shorter training regime than full-scale models (“50 epochs instead of 400”, cf. page 8). The methodology may not uncover longer-term stability issues or provide sufficiently representative results for convergence characteristics. Moreover, for Table 2 ("Component Contributions"; values not explicitly shown in the main text), further detail and numbers should be presented.

**Questions:**

Please provide responses to the issues raised in my Weaknesses section.

---

> ### Author Response · Authors · 2025-11-20
>
> We sincerely thank the reviewer for the constructive comments and for acknowledging the **"excellent soundness"** and **"strong empirical results"** of our work.
>
> We address your specific concerns below:
>
> > **Q1: "Limited Discussion of Generalization and Failure Cases... questions about the robustness and transferability.”**
>
> **Response:**
> We thank the reviewer for this valuable suggestion. To comprehensively address the concerns regarding robustness, scalability, and limitations, we have conducted **three additional experiments/analyses**:
>
> 1. **Scaling to Higher Resolution:** Addressing the specific request for higher resolutions, we extended SphereAR to ImageNet $512 \times 512$ (see **Appendix E**). It achieves an **FID of 1.67**, outperforming previous baselines.
>
> 2. **Scaling Model Size:** To further verify the robustness and scalability of our architecture, we trained a larger model, **SphereAR-H (943M)**, to match MAR-H. As reported in Table 1, SphereAR-H achieves an **FID of 1.34**, setting a new SOTA.
>
> 3. **Failure Case Analysis:** **Following the reviewer's specific suggestion for qualitative exploration**, we have added a detailed analysis of failure modes in **Appendix I**. Similar to existing methods, SphereAR can produce artifacts in complex classes (e.g., text rendering or faces). However, comparisons show that **SphereAR-L consistently produces visually superior results with fewer artifacts than the baseline MAR-L** even in these challenging scenarios.
>
> **Regarding generalization to other datasets:**
> We agree that extending SphereAR to domains beyond ImageNet is a valuable direction. However, ImageNet-1K is the **established standard benchmark** for validating the performance of image generation (as seen in foundational works like DiT, VAR, and MAR). Given this precedent and the substantial computational resources required for training on new diverse datasets, we focused our current evaluation on this standard.
>
> We will release our code and model weights to enable the community to explore SphereAR's transferability to diverse tasks and datasets.
>
> > **Q2: “Empirical Focus Narrowed to FID and Class-Conditional Generation”**
>
> **Response:**
> We fully agree with the reviewer that metrics like FID and IS have limitations in fully capturing semantic consistency or fine-grained diversity.
>
> However, we emphasize three points regarding our evaluation choices:
>
> 1. **De-facto Standard:** FID and IS remain the established quantitative benchmarks for ImageNet generation. They are indispensable for providing a **fair, apples-to-apples comparison** with prior state-of-the-art methods (e.g., DiT, MAR, VAR), all of which rely primarily on these metrics.
>
> 2. **Lack of Consensus on Alternative Metrics:** Currently, the community has not yet converged on a universally accepted alternative metric that better decouples semantic consistency from image quality for class-conditional generation. **Notably, the vast majority of our baselines do not report such metrics, making direct comparison unfeasible.**
>
> 3. **Qualitative Supplement:** To complement these quantitative metrics and address their limitations, **we have explicitly adopted the reviewer’s suggestion** to include a **Failure Case Analysis** (Appendix I) alongside extensive qualitative samples (Appendix E). These allow for a visual inspection of semantic faithfulness and failure modes that raw numbers might miss.
>
> We will release our code and pretrained models to facilitate future evaluations using more advanced semantics-aware metrics as they become available.

---

> ### Author Response · Authors · 2025-11-20
>
> > **Q3: “Ablation Duration and Robustness... methodology may not uncover longer-term stability issues.”**
>
> **Response:**
> We thank the reviewer for noting this implementation detail. We justify our choice of a 50-epoch ablation regime based on three key factors:
>
> 1. **High Convergence Efficiency:** As demonstrated in **Appendix F**, SphereAR exhibits extremely high training convergence efficiency (converging $\sim$5x faster than masked-generation models). Therefore, 50 epochs for SphereAR provide a much stronger training signal than 50 epochs for diffusion/masked-generation models, making this duration sufficient to reliably distinguish the relative effectiveness of different components.
> 2. **Long-term Stability Verification:** While ablations were short, our main models (SphereAR-L/H) were successfully trained for the full 400 epochs without any instability. This serves as direct empirical evidence that the proposed S-VAE and normalization mechanisms remain stable over long-term training regimes.
> 3. **Common Practice:** Using shorter schedules for ablation is a standard trade-off in the resource-intensive generative modeling literature. For instance, **LlamaGen** also uses 50 epochs for ablations, **RandAR** uses ~83 epochs (100k steps), and **LatentLM** uses 40 epochs.
>
> Given the limited computational resources and the need to compare 15+ model variants, we believe this setting offers the best balance between feasibility and statistical representativeness.
>
> > **Q4: “Table 2 ("Component Contributions"; values not explicitly shown in the main text)”**
>
> **Response:**
> We thank the reviewer for pointing out this omission. We have revised Section 4.3 in the main text to **explicitly incorporate the numerical values and detailed analysis** corresponding to Table 2.

---

> > ### Author Response · Authors · 2025-12-01
> > **Additional Ablation Results with 200 Epochs**
> >
> > To further demonstrate the robustness of our ablation study, we extended the experiments in Table 2 from 50 to 200 epochs.
> >
> > As summarized in the table below, the relative performance ranking remains perfectly consistent in the longer training regime. Specifically, the Spherical posterior continues to outperform the Gaussian baselines by a clear margin at 200 epochs (FID 2.13 vs 2.33). These results empirically validate that our 50-epoch ablation is a reliable proxy for long-term model performance. We hope this additional experiment addresses your concern regarding robustness of our ablation study.
> >
> > | Norm. on VAE Dec. | Norm. on AR | Posterior | FID (50 ep) | FID (200 ep) | IS (50 ep) | IS (200 ep) |
> > | :---: | :---: | :---: | :---: | :---: | :---: | :---: |
> > | ✗ | ✗ | Gaussian | 2.97 | 2.53 | 240.2 | 248.3 |
> > | ✓ | ✗ | Gaussian | 2.89 | 2.46 | 254.3 | 258.4 |
> > | ✓ | ✓ | Gaussian | 2.68 | 2.33 | 257.3 | 265.8 |
> > | ✓ | ✓ | **Spherical** | **2.52** | **2.13** | **258.4** | **268.7** |

---

### Official Review · Reviewer_hP4t · 2025-11-01

**Soundness:** 3
**Presentation:** 3
**Contribution:** 3
**Rating:** 6
**Confidence:** 3

**Summary:**

This paper points out that current autoregressive image generative models based on continuous tokens often performs unsatisfyingly. To tackle this problem, this paper proposes SphereAR. Specifically, the authors analyzes the importance of scale-invariant tokens for AR generation with continuous tokens. Based on this, the authors propose to replace the original VAE with S-VAE, which parameterizes each token on a unit sphere with the same norm. Then the authors combine the S-VAE with an autoregressive transformer and use the diffusion loss to train the model. Experiments show that SphereAR outperforms all mainstream AR and masked generation models with the same model parameter size.

**Strengths:**

1. The motivation is clear, and the analysis regarding the importance of keeping the scale of tokens the same is reasonable.
2. Using S-VAE is a straightforward yet effective modification.
3. The experimental results are impressive. The design of the ablation study is also reasonable.

**Weaknesses:**

Overall, this paper is quite good. I only have the following minor concern.
1. Since the authors use an AR architecture of transformer, which is different from MAR, it would be better to include the performance of combining this architecture with a standard VAE as a baseline, which is a fair comparison.

**Questions:**

1. The author mentions the different behaviors of masked models and autoregressive (AR) models when both use continuous tokens in Sec.3. I am curious about the underlying reason for this difference. In other words, what fundamentally distinguishes masked models from AR models with continuous token?

---

> ### Author Response · Authors · 2025-11-20
>
> We sincerely thank the reviewer for the positive assessment and the acknowledgment of our clear motivation and impressive results.
>
> We address your specific suggestions below:
>
> > **Q1: "it would be better to include the performance of combining this architecture with a standard VAE as a baseline"**
>
> **Response:**
> We thank the reviewer for this constructive suggestion to ensure a fair comparison. We have included this exact baseline in **Appendix C**.
>
> Specifically, we trained a model combining our AR architecture with the standard KL-regularized Gaussian VAE (used in MAR). The results show that replacing the standard VAE with our **S-VAE** yields a substantial performance leap: **FID improves from 4.54 to 2.52**, and IS increases from 241.6 to 258.4.
>
> This ablation confirms that the performance gain comes primarily from the **constant-norm, directional latents** provided by S-VAE, rather than the AR architecture itself.
>
> > **Q2: “what fundamentally distinguishes masked models from AR models with continuous token?”**
>
> **Response:**
> The fundamental distinction lies in two key aspects: the **number of autoregressive steps** and the **generation order**.
>
> 1. **Number of Steps & Error Accumulation:**
>    - **Standard AR:** Must generate tokens sequentially one by one (e.g., 256 steps for 256 tokens). Since CFG rescaling induces a slight norm shift at *each* step, these errors accumulate deeply over the full sequence length, leading to significant variance collapse or drift.
>    - **Masked Models (e.g., MAR):** Leverage parallel token sampling. They generate tokens in groups (chunks), typically requiring far fewer steps (e.g., only 64 steps for 256 tokens). Consequently, the error accumulation caused by CFG rescaling is **significantly less pronounced** compared to standard AR.
>
> 2. **Generation Order:**
>    - **Masked Models:** Typically generate tokens in a **random order**. As evidenced by prior works like RandAR and RAR (and shown in Table 1), random-order modeling can inherently improve generation performance compared to fixed raster ordering.
>    - **Standard AR:** Typically follows a fixed **raster order**.
>    - **Compatibility:** Importantly, the "random order" strategy is **orthogonal** to our SphereAR contribution. SphereAR focuses on the geometric stability of the latent space; it could theoretically be combined with random-order generation to achieve even further performance gains.

---

### Author Response · Authors · 2025-11-20
**Summary of Revisions & Additional Experiments**

We sincerely thank all reviewers for their constructive comments and valuable time. Based on your suggestions, we have revised the manuscript and conducted additional experiments to further validate SphereAR.

In particular, we have made the following key revisions:

- **Scaled-up Model (Table 1):** We introduced a large model, **SphereAR-H (943M)**, to match the capacity of MAR-H. It achieves a new SOTA FID of **1.34**.
- **Higher Resolution (Appendix E):** We extended SphereAR to ImageNet $512 \times 512$ generation, achieving an FID of **1.67**.
- **New Baseline Ablation (Appendix C):** We added an ablation using the standard VAE (from MAR) within our AR architecture to isolate the gain from S-VAE (suggested by Reviewer **hP4t**).
- **Training Efficiency Analysis (Appendix F):** We provided a wall-clock convergence efficiency comparison against MAR and VAR (suggested by Reviewer **LHMS**).
- **Inference Speed Benchmark (Appendix G):** We added a detailed quantitative inference speed comparison (suggested by Reviewer **xjkB**).
- **High CFG Robustness Analysis (Appendix H):** We analyzed performance degradation under aggressive CFG scales (suggested by Reviewer **xjkB**).
- **Failure Case Analysis (Appendix I):** We included a qualitative analysis of failure modes and limitations (suggested by Reviewer **Li3p**).

The revised manuscript has been uploaded. We hope these additional results address your concerns.

---

### Meta-Review · Area_Chair_w65W · 2026-01-17

**Summary:**

This paper targets a key failure mode in continuous-token autoregressive image generation: heterogeneous latent scale that can amplify during multi-step AR decoding (especially with CFG), leading to instability/variance collapse. The authors propose SphereAR, which leverages hyperspherical latents end-to-end, i.e., all AR inputs/outputs are projected onto a fixed-radius hypersphere. Empirically, SphereAR reports very strong ImageNet results.

Across reviews, the paper is viewed as technically sound with clear theoretical motivation and unusually strong ImageNet results. The rebuttal materially strengthens the empirical story (scale-up to FID 1.34, higher-res 512×512, longer ablations, added runtime/efficiency and robustness analyses, and the key “standard VAE + same AR” baseline). AC recommends acceptance.

**Reviewer Concerns:**

Substantially addressed in rebuttal:
- Fair baseline / attribution of gains (hP4t): Authors added the requested standard VAE + their AR architecture baseline and show a large gap vs SphereAR (e.g., FID 4.54 → 2.52 when swapping in S-VAE), strengthening causal attribution to the hyperspherical latents rather than architecture differences.
- Generalization to higher resolution (Li3p, xjkB): Added ImageNet 512×512 experiment (FID 1.67) and emphasized scale-up to SphereAR-H (943M) FID 1.34.
- Ablation robustness / short training regime (Li3p): Extended ablations to 200 epochs and provided the corresponding table, showing consistent rankings and a clearer margin for spherical posterior (e.g., FID 2.13 vs 2.33 at 200 epochs).
- Runtime / efficiency and CFG robustness (xjkB, LHMS): Added wall-clock efficiency discussion (Appendix F), inference speed benchmarks (Appendix G), and high-CFG robustness analysis (Appendix H).

Partially addressed / still outstanding:
- Breadth beyond ImageNet class-conditional: Reviewers explicitly noted the evaluation is limited to ImageNet class-conditional generation and asked about transfer to other datasets / text-to-image; rebuttal provides arguments and partial evidence (512×512, scale-up), but still no non-ImageNet / non-class-conditional results.
- Metrics beyond FID/IS: Authors acknowledge limitations and add qualitative/failure analysis, but the quantitative story remains centered on standard metrics.
- Sampling speed remains a structural concern: Runtime comparisons help, but SphereAR is still AR with a diffusion head, so speed competitiveness will depend on chosen diffusion steps and future few-step improvements.

**Reviewer Scores:**

- Reviewer hP4t (original: 6): likely remain 6 or improve to 7.
- Reviewer Li3p (original: 4): possibly 6. The rebuttal directly targets their three main weaknesses—higher-res generalization, failure-case discussion, and ablation length/robustness—with concrete new experiments and clearer reporting. The main remaining reason to stay cautious is limited evaluation beyond ImageNet and limited non-FID metrics.
- Reviewer xjkB (original: 6): likely stay at 6
- Reviewer LHMS (original: 8): likely no change.

---

### Decision · Program_Chairs · 2026-01-26

Accept (Poster)